# Molecular dynamics study on the strengthening behavior of Delta and Omicron SARS-CoV-2 spike RBD improved receptor-binding affinity

**Kanchanok Kodchakorn⬥, Prachya Kongtawelert***

Thailand Excellence Center for Tissue Engineering and Stem Cells, Department of Biochemistry, Faculty of Medicine, Chiang Mai University, Chiang Mai, Thailand

* prachya.k@cmu.ac.th

## Abstract

The COVID-19 pandemic caused by a virus that can be transmitted from human to human via air droplets has changed the quality of life and economic systems all over the world. The viral DNA has mutated naturally over time leading to the diversity of coronavirus victims which has posed a serious threat to human security on a massive scale. The current variants have developed in a dominant way and are considered "Variants of Concern" by the World Health Organization (WHO). In this work, Kappa (B.1.617.1), Delta (B.1.617.2), and Omicron (B.1.1.529) variants were obtained to evaluate whether naturally occurring mutations have strengthened viral infectivity. We apply reliable *in silico* structural dynamics and energetic frameworks of the mutated S-RBD protein for ACE2-binding to analyze and compare the structural information related to the wild-type. In particular, the hotspot residues at Q493, Q498, and N501 on the S-RBD protein were determined as contributing factors to the employment stability of the relevant binding interface. The L452R mutation induces an increment of the hydrogen bonds formed by changing the Q493 environment for ACE2 binding. Moreover, the Q493K exchange in Omicron enables the formation of two additional salt bridges, leading to a strong binding affinity by increased electrostatic interaction energy. These results could be used in proposing concrete informative data for a structure-based design engaged in finding better therapeutics against novel variants.

## Introduction

The severe acute respiratory syndrome coronavirus 2 (SARS-CoV-2) is driven by newly emerging variants nearly two years into the pandemic worldwide that has deeply affected the entire population, pushing young health professionals into new careers outside the medical field. A series of coronavirus variants has emerged in different countries, some of which have caused significant outbreaks. The variants of Alpha [1], and more recently Delta [2], have had the greatest global reach. The variants of Beta [3], Gamma [4], and Lambda [5], did not

**Data Availability Statement:** All relevant data are within the paper and its Supporting Information files.

**Funding:** This work was supported by grants from the Post-Doctoral Fellowship, grant number: R000030567, by CMU Presidential Scholarship, Chiang Mai University, Chiang Mai, Thailand. However, the founder had no role in study design, data collection and analysis, decision to publish, or preparation of this manuscript.

**Competing interests:** The authors have declared that no competing interests exist.

become dominant mutations in most regions of the world, although they did cause substantial outbreaks in Southern Africa (SA). Nowadays, the growing presence of a novel dominant strain of the virus is a reminder the pandemic is far from over. Meanwhile, the personal choices made by people during the pandemic, especially concerning vaccinations, have shaken the foundations of health systems.

On November 26, 2021, the WHO designated one recent variant B.1.1.529, called "Omicron", a "Variant of Concern (VOC)" with high potential to spread rapidly across the world [6, 7]. This variant is spreading significantly faster than the Delta variant and has recently appeared in SA [8]. At the same time, Omicron is quickly overtaking Delta to be the dominant strain of SARS-CoV-2 in the US and many other countries [8–10]. Both variants are complicating the work of the medical professionals who are responsible for assisting patients with breathing challenges. Thus, it is a time of year when numerous respiratory illnesses emerge. The rapid emergence of Omicron in the background of the immune response implies that the virus may have evolved to escape neutralizing antibody responses by Beta-specific serum [11]. Omicron is a highly divergent variant with a high number of mutations (thirty mutations in the spike (S) protein, 15 of which occur in the receptor-binding domain (RBD), as well as three small deletions and one minor insertion), some of which may be associated with humoral immune escape potential and higher transmissibility. Particular potential hotspot residues for the mutations are the S-RBD region as the site of ACE2 binding that dramatically improves RBD-ACE2 binding affinity, causing fast dissemination in human populations [12].

The S-RBD protein is an essential part of SARS-CoV-2 as it mediates interaction with human cells and is the target for therapeutic neutralizing antibodies [13, 14]. Several studies have proposed the mechanism of the ACE2-bound S-RBD binding in the viral entry process [15, 16]. The mutation's effects on S-RBD binding affinity and viral expression are found to correlate with an amino acid residue as the key hotspots clustering at the binding interface and has been well documented [17–19]. Indeed, a bacterial surface-displayed variety of the mutated S-RBD selected for tighter ACE2 affinity presents in the variants of concern N501Y (Alpha, Beta, and Gamma) and E484K (Beta and Delta), and multiple other variants, such as S477N [20]. The variant harbors two mutations within the RBD region (L452R and E484Q), the region responsible for the viral entry [21]. Furthermore, the Q498R exchange has shown a major contribution to ACE2 binding (in epistasis with N501Y). A modest change in affinity might cause a significant increase in the infection rate. As mentioned above, Omicron is found to harbor a common feature of N501Y mutation in the S-RBD protein that is not present in Delta, indicating that this new variant of Omicron provides different affinity enhancing mutations that may arise [20, 22], and is also more transmissible or severe than the Delta variant. The side chain of the Q498R mutation forms two additional hydrogen bonds with Q42 and D38 from ACE2, and the N501Y mutation forms extensive packing interactions with the ACE2 residues Y41 and K353 [23]. Compared to Omicron, the Delta variant carries a characteristic L452R mutation, which is the strong binding interaction of the S-RBD protein described for ACE2 binding [24]; however, this mutation would not clash with the Y102 epitope from the heavy chain of the antibody, leading to its loss of potency against the Delta variant [25]. Both the Delta and Omicron variants share a common T478K exchange, while a low affinity for neutralizing antibodies at this position was observed for Delta [26, 27]. Thus, further surveillance, diagnosis, evaluation, and treatment of the mutated SARS-CoV-2 strains are necessary.

Based on the field of computational drug discovery of protein-protein and protein-ligand interactions, we proposed detailed structural behaviors at the potential binding interface that would allow us to gain a better understanding of the mutated molecular effects on the spike protein of SARS-CoV-2 with its cellular receptor [28–33]. The main purpose of the present

work was to compare the structural information and energetic affinity between the Wuhan Hu 1 wild-type and the mutated (Kappa, Delta, and Omicron) variants in complex with ACE2 protein by using a variety of computational techniques. Despite the computational demands, the binding free energy approach, such as MM-PBSA method, is one of the most successful and precise *in silico* techniques for accurate prediction of the ligand selectivity [34], protein-protein interactions [35], and protein stability [36]. It has been shown that the major changes in the binding affinity between the S-RBD and ACE2 protein occurred at various residue positions that almost mutated in the S-RBD region. Moreover, most of the remaining residue exchanges at the binding interface did not generally affect binding with the receptor but may instead change epitopes for the neutralizing antibody response [37, 38].

## Materials and methods

### Mutation variants and structure preparation

The FASTA sequence of the SARS-CoV-2 of Wuhan-Hu-1 (wild-type) was obtained from Uniport 13. (GenBank: QHD43415.1) [39]. The Kappa (GenBank: MZ157006.1), Delta (GenBank: QWK65230.1) and Omicron (R40B60 BHP 3321001247/2021) variants were obtained from GSAID database. The translated sequence was performed to collect the omicron spike protein. Clustal-X2, a bioinformatics program, was used to align the wild-type sequence with variants of delta and omicron sequences. The box shade application in each sequence alignment of the SARS-CoV-2 S-RBD variants: Kappa (B.1.617.1), Delta (B.1.617.2), and Omicron (B.1.1.529), in comparison with the wild-type (WT) was showed in Fig 1. The starting structure for the wild-type SARS-CoV-2 S-RBD protein in complex with ACE2 were obtained from the RCSB Protein Data Bank (PDB ID: 6M0J) [40]. We construct a various type of the mutated S-RBD protein according to Fig 1 by direct replacing *in silico* mutated amino acids of each mutation in the S-RBD region (438–505, red color) of the wild-type S-RBD in an experimental PDB structure before proceeding to the next MD simulations step.

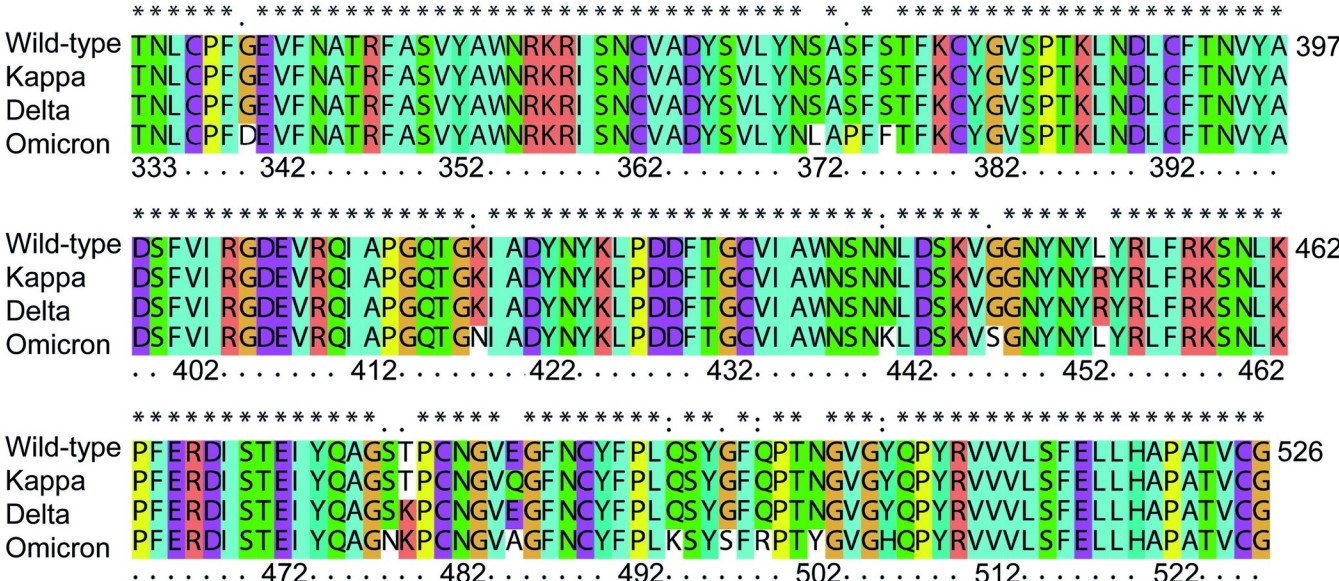

**Fig 1. Amino acid sequence alignment in each SARS-CoV-2 S-RBD variant; wild-type, Kappa (B.1.617.1), Delta (B.1.617.2), and Omicron (B.1.1.529).** The ACE2 receptor binding domain (RBD, or spike RBD region) are located in residues 438–505.

## Molecular dynamics simulation and binding free energy analysis

All molecular dynamics (MD) simulations were performed by PMEMD.CUDA [41, 42] from AMBER 18 suite of programs [43] on NVIDIA Geforce GTX-1070 Ti graphic card for speeding up the simulation times. Each mutation complex under periodic boundary condition was solvated in a cubic box of TIP3P water molecules extending to 10 Å along each direction from the complex model, and $Na^+$ ions were added as neutralizing counterions. The cutoff distance was kept to 12 Å in order to compute the non-bonded interactions. The AMBER ff14SB force field parameters were used to apply the description of the complex characterization. The long-range electrostatic were treated using the particle mesh Ewald (PME) method [44]. The SHAKE algorithm and Langevin dynamics were applied to constrain the bonds that involved hydrogen atoms and to control the temperature.

The mutated three-dimensional models were first minimized using single point energy calculation to relax and remove overlapping atoms. The first step was to allow, out of 10,000 iterations, only water molecules to move. In the second step, each of the 10,000 iterations, hydrogen and protein side chains were relaxed, in a fixed order. Finally, 20,000 iterations were calculated with the restriction-free system. After that, the optimized MD systems were carried out gradually heating (H) phase NVT (constant number of atoms, volume, and temperature) ensemble with the fixed protein atoms over 100 ps from 0 K to 310.15 K by using a force constant of 10 kcal mol$^{-1}$ Å$^{-2}$. This was followed by 1,000 ps of equilibration-1 (Eq 1) phase NVT-MD at 310.15 K at a force constant of 5.0 kcal mol$^{-1}$ Å$^{-2}$. Then, 10,000 ps without any constrained forces of equilibration-2 (Eq 2) phase NPT (NPT; constant number of atoms, pressure, and temperature) were performed for each fully flexible equilibrium system at the same temperature and 1 atm pressure. The density of each system was about 1.0 g cm$^3$. At the end of the production period, all memory the initial configuration was intentionally lost. This was done so that the final results would not depend on the initial configuration. Finally, 50 ns of unrestrained production (Prod) phase MD simulation were applied in the NVT ensemble at a constant temperature of 310.15 K. The time step of 2 fs was set and the trajectory was recorded every 0.2 ps. Trajectory analyses (root mean square deviation and fluctuation, dynamic cross-correlation, hydrogen bond) were carried out using CPPTRAJ module [45] from Amber 18 program from all 50 ns trajectories data. The structural images were presented using DS software.

Amber molecular mechanics Poisson–Boltzmann surface area (MM-PBSA) [46] approach was used to calculate the binding free energy of S-RBD to ACE2 using the snapshots extracted from MD simulation trajectories. The 2500 numbers of structural frequencies were used to extract the structures from last 50 ns trajectories data of the Prod phase MD simulations. The 1000 snapshots were collected from the trajectory data to calculate the binding free energy by the MM-PBSA procedure with the grid size of 0.5 Å. The binding free energy ($\Delta G_{binding}$, kcal mol$^{-1}$) was calculated for each molecular species (each ACE2-bound S-RBD variant complex) in the energetic framework of the MM-PBSA method, and the $\Delta G_{binding}$ was computed using Eq 1:

$$\begin{aligned}\Delta G_{binding} &= G_{S-RBD/ACE2} - (G_{S-RBD} + G_{ACE2}) \\ &= \Delta E_{vdW} + \Delta E_{EEL} + \Delta G_{SOL} - T\Delta S = \Delta H - T\Delta S\end{aligned} \quad (1)$$

in which $\Delta E_{vdW}$ and $\Delta E_{EEL}$ refer to the van der Waals and electrostatic contribution as calculated by the molecular mechanics force field, and $\Delta G_{SOL}$ represent the solvation free energy. The internal dielectric constant was set to the values of 2, 3 and 9 for nonpolar, polar, and charged residues, respectively.

**Computational alanine-scanning mutagenesis analysis.** For alanine-scanning mutagenesis, the role of the protein-protein interface key residues was studied by performing computational alanine scanning calculation [47]. Each mutant structure was prepared by Amber 18 software. The mutated amino acid of S-RBD mutant system was directly replaced in the wild-type model of SARS-CoV-2 S-RBD protein (pdb file). Here, this mutation involves all side chain atoms except the beta-carbon (CB) were removed and also their corresponding information (name, number, and coordinates) from the pdb files. Finally, change the residue name to "ALA" for all remaining atoms of the mutant residue. Other mutations will be able to follow similar procedures where the group of atoms after CB but before the carbonyl-corbon (C) may be removed from the pdb file. It can be noted that only one mutation can be measured during a single calculation. Then, the corresponding topology files in each mutant structure was built and calculated the binding free energy by MM-PBSA method. The $\Delta G_{binding}$ of each mutant structural system, in which each key residue was replaced by alanine (A) by truncating the mutated residue at the Cγ-atom, was calculated with the MMPBSA method. The difference in the binding free energy between the wild-type ($\Delta G_{wild-type}$) protein and its alanine mutant ($\Delta G_{alanine}$) counterpart, $\Delta\Delta G_{binding}$ (kcal mol$^{-1}$), is given by Eq 2:

$$\Delta\Delta G_{binding} = \Delta G_{wild-type} - \Delta G_{alanine} \tag{2}$$

A negative $\Delta\Delta G$ values indicate unfavorable substitution for alanine in the relevant position, meanwhile, indicating a favorable contribution for the wild-type residue in that position.

**Hydrogen bond analysis.** To search for hydrogen bonds formation, the number of hydrogen bonds present at each frame will be determined, hydrogen bond donors and acceptors of heavy atoms, will be counted using *hbond* command in CPPTRAJ module which were printed with the *avgout* format [48]. The distance cutoff for hydrogen bonds (acceptor to donor heavy atom) was set at 3.5 Å, while angel cutoff for hydrogen bonds was performed at 135˚. Hydrogen bond occupancy for the residue pairs which are calculated by averaging over production phase NPT-MD simulations were written in term of Eq 3:

$$\text{Frac} = \frac{\text{Frames of HB cutoff } 3.5\text{Å from acceptor to donor atom from MD simulations}}{\text{Total Frames from MD simulations}} \tag{3}$$

where Frames is the number of each coordinate structure that the observed hydrogen bond is present.

**Dynamic cross-correlation matrix analysis.** Dynamic movement and correlations between the Cα–atoms positions of SARS-CoV-2 spike RBD and ACE2 protein over the simulation period were calculated by dynamic cross-correlation matrix (DCCM) analysis. DCCM was performed using CPPTRAJ module of the AMBER 18 suite. DCCM diagrams are displayed as a color-coded matrix of Pearson correlation coefficients. Where there is a highly-correlated motion, the movement towards the same direction between the residue pairs shows a positive value in the color range from light green to deep red (+1); while in anti-correlated motions, the movement shows a negative value in the color range from gray to royal blue (-1) [49]. The diagonal square relates to the correlation of a residue with itself, *i.e.*, only a region remarked to have highly-positive values (red). The potential amino acids in binding site of S-RBD protein were clarified at residue number of 438–505.

**Principal component analysis.** To evaluate the displacement of atoms and conformational dynamics of a protein complex, principal component analysis (PCA) was performed and analyzed using a covariance-matrix-based approach [50, 51]. The elements of the positional covariance matrix C were obtained based on the following Eq 4:

$$C_{ij} = \langle (x_i - \langle x_i \rangle) \times (x_j - \langle x_j \rangle) \rangle \tag{4}$$

where $x_i$ and $x_j$ are the instant coordinates of the $i^{th}$ and $j^{th}$ Cα-atoms of the systems for using in building matrix $C$, while $\langle x_i \rangle$ and $\langle x_j \rangle$ refers to an ensemble average. The averaged values are computed over the Prod phase MD simulations after superimposition on a reference structure using the CPPTRAJ module of the AMBER 18 package, solvent water molecules and neutralizing ions added by the Leap module are stripped prior to MD trajectory generation. PCA was performed for Cα-atoms on 10,000 snapshots each. PC1 and PC2, which represent the first two principal components, are created from the trajectories averaged from the wild-type and the variant system. The trajectories were analyzed for the relative motions about their center of masses.

## Results and discussion

### Multiple alignment and physical parameters of spike-RBD variant proteins

Many mutations in the S-RBD region were found in the Omicron variant compared to the others according to the multiple sequence alignment, as illustrated in Fig 1. This indicates that Omicron may be immunologically resistant to the neutralizing response. The mutated spike protein of Omicron includes more than 30 mutations, half of which are located in the RBD regions. The S-RBD loop at the K440-Y505 positions was observed as a viral determinant for ACE2 binding [52, 53]. Furthermore, the residue of T478 is a common mutation seen in both the Delta and Omicron variants.

Initially, the primary structure analysis shared by all SARS-CoV-2 variants reveals that there is an increase in the following amino acid compositions in the S-RBD of the Omicron variant compared to the Delta (Table 1): arginine (R), lysine (K), histidine (H), and glutamic acid (E), suggesting Omicron has more amino acids with electrostatically charged side chains that can contribute to the salt bridge interaction of the ACE2 protein [54]. The higher percentage residue composition of phenylalanine (F), leucine (L), and alanine (A) was observed in the

**Table 1. Amino acid percentage composition of the mutated SARS-CoV-2 variants with reference to the wild-type in the residue range of 333–526 positions and contact to the RBD protein.**

| Residue types | Wild-type | WT-RBD | Kappa | Kappa-RBD | Delta | Delta-RBD | Omicron | Omicron-RBD |
|---|---|---|---|---|---|---|---|---|
| Alanine (A) | 6.2 | 1.3 | 6.2 | 1.3 | 6.2 | 1.3 | 6.4 | 2.0 |
| Arginine (R) | 4.6 | 5.3 | 4.9 | 5.8 | 4.9 | 5.8 | 4.9 | 6.0 |
| Asparagine (N) | 8.8 | 12.0 | 8.8 | 12.0 | 8.8 | 12.0 | 8.8 | 11.3 |
| Aspartic acid (D) | 4.6 | 2.7 | 4.6 | 2.7 | 4.6 | 2.7 | 4.9 | 2.7 |
| Cysteine (C) | 4.1 | 2.7 | 4.1 | 2.7 | 4.1 | 2.7 | 4.1 | 2.7 |
| Glutamine (Q) | 3.1 | 5.3 | 3.4 | 6.0 | 3.1 | 5.3 | 2.6 | 4.0 |
| Glutamic acid (E) | 3.1 | 4.0 | 2.8 | 3.3 | 3.1 | 4.0 | 2.8 | 3.3 |
| Glycine (G) | 7.7 | 10.7 | 7.7 | 10.7 | 7.7 | 10.7 | 7.0 | 9.3 |
| Histidine (H) | 0.5 | 0.0 | 0.5 | 0.0 | 0.5 | 0.0 | 0.8 | 0.7 |
| Isoleucine (I) | 3.6 | 2.7 | 3.6 | 2.7 | 3.6 | 2.7 | 3.6 | 2.7 |
| Leucine (L) | 6.7 | 6.7 | 6.4 | 6.0 | 6.4 | 6.0 | 7.0 | 6.7 |
| Lysine (K) | 4.1 | 4.0 | 4.1 | 4.0 | 4.4 | 4.7 | 4.6 | 6.0 |
| Phenylalanine (F) | 7.2 | 6.7 | 7.2 | 6.7 | 7.2 | 6.7 | 7.5 | 6.7 |
| Proline (P) | 5.2 | 6.7 | 5.2 | 6.7 | 5.2 | 6.7 | 5.4 | 6.7 |
| Serine (S) | 7.7 | 8.0 | 7.7 | 8.0 | 7.7 | 8.0 | 7.2 | 8.7 |
| Threonine (T) | 5.7 | 4.0 | 5.7 | 4.0 | 5.4 | 3.3 | 5.4 | 3.3 |
| Tryptophan (W) | 1.0 | 0.0 | 1.0 | 0.0 | 1.0 | 0.0 | 1.0 | 0.0 |
| Tyrosine (Y) | 7.7 | 10.7 | 7.7 | 10.7 | 7.7 | 10.7 | 7.7 | 10.7 |
| Valine (V) | 8.2 | 6.7 | 8.2 | 6.7 | 8.2 | 6.7 | 8.2 | 6.7 |

whole protein of Omicron compared to other variants. These non-polar residues are located inside the protein backbone core and are inaccessible to the environmental solvent [55]. Additionally, the residue composition of Omicron is low in polar uncharged side chains, such as asparagine (N) and glutamine (Q), when compared to the Delta mutation.

## Mutational scanning and energetic analysis of the wild-type S-RBD residues at the binding interface with ACE2

According to recent X-ray results [40, 16, 56, 57], we found that the simulated structure of the ACE2-bound SARS-CoV-2 S-RBD protein preserves the same binding interface information, which does not induce any conformational change in the receptor-binding site [24, 58–61]. One essential feature at the binding interface hotspot between S-RBD and ACE2 proteins is the number of hydrophilic residue interactions [16, 40], which are conserved in the corresponding MD simulation calculation. Certainly, the hydrogen-bonding hotspots and two salt bridges (K31-E35 and K353-D38) in the present work stably populate S-RBD binding with the ACE2 receptor [58, 62–64]. Here, the energetic contributions of wild-type S-RBD residues were firstly analyzed over the production phase MD simulations for understanding the mapping key population hotspot in the SARS-CoV-2 S-RBD wild-type protein. That is to say, the atomistic structural characterisation of the binding interface residues could better quantify the binding energetic differences between the spike protein RBD variants and its cellular ACE2 receptor, providing fundamental information about the design of structure-based vaccines and/or neutralizing antibodies development.

To provide the qualification of the binding energetic characterization and why they are causing concern in the S-RBD variants, we employed the combined description for all topical protein-protein interface interactions along with the corresponding energetic quantification discussed in detail by an alanine scanning calculation through the MM-PBSA method on the binding free energy of the mutated S-RBD complex with ACE2. Each mutant structure in the pdb file was prepared by Amber 18 software as listed in S1 Table. For instance, mutated A498 residue of the S-RBD mutant system was directly replaced in Q498 of the wild-type model of the SARS-CoV-2 S-RBD protein (pdb file). Other mutations will be able to follow similar procedures in each mutation during a single calculation. Then, the corresponding topology files in each mutant structure were built and the binding free energy calculated by MM-PBSA method for comparison to the wild-type MD simulation as the trajectories to be analyzed. As a result, the significant relative binding free energy ($\Delta\Delta G_{binding}$, kcal mol$^{-1}$) of each mutant was reported, demonstrating the relative effect the mutation has on the $\Delta G_{binding}$ of binding for the complex as shown in Fig 2. It can be noted that the negative values of the $\Delta\Delta G_{binding}$ obtained from the computational alanine scanning mutagenesis indicate unfavorable substitutions for alanine in the relevant position. Meanwhile, it also indicates a favorable contribution for the wild-type residue in that position and vice versa.

The strongest binding energies of hotspot residues are located in the central segment of the interface, $i.e.$, N487, Q493, and Q498. In other words, a large and comparable loss of binding energy is observed, which forms favorable hydrophilic contacts with ACE2. N487 contributes to ACE2 binding with two interface-anchoring Q24 and Y83 hydrogen-bonding interactions (Fig 3A). Alanine substitution at the N487 position is accompanied by a ~6.0 kcal mol$^{-1}$ loss in binding free energy ($\Delta\Delta G_{N487A}$ = −5.92 ± 2.55 kcal mol$^{-1}$). At the Q493 position, one intermolecular hydrogen bond with K31 and E35 of the ACE2 protein and one intramolecular interaction with the S494 side chain (occupancy 22.35%, distance$_{avg}$ = 3.082 Å) were found to form at the binding interface (Fig 3B). Likewise, Q498 of the S-RBD protein establishes an extensive network of favorable interactions across the ACE2 binding interface with K353-D38 salt

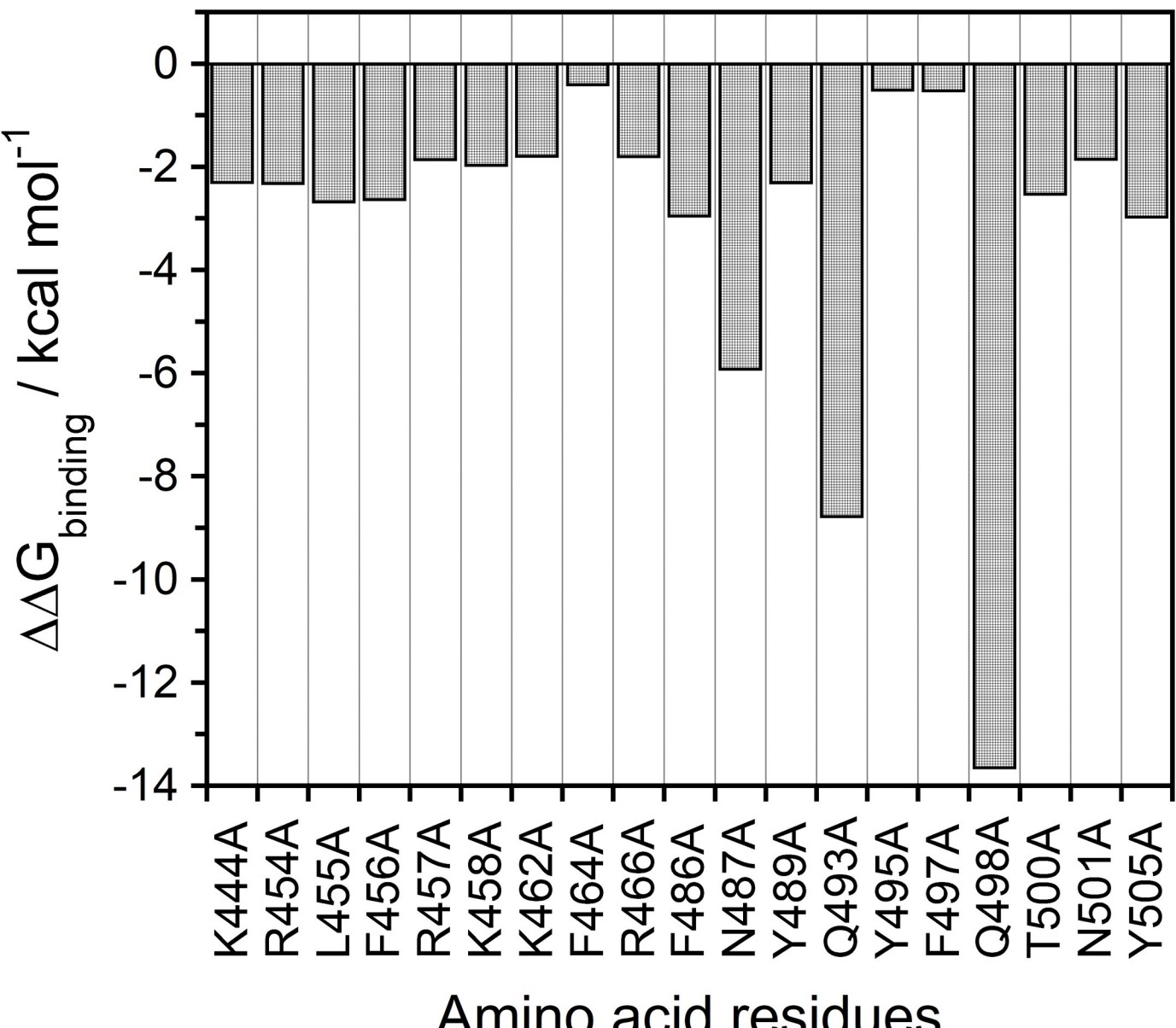

**Fig 2. An alanine scanning of spike-RBD binding region in the wild-type SARS-CoV-2 structure over the production phase MD simulations.** The binding energy changes ($\Delta\Delta G_{binding}$, kcal mol-1) are reported that demonstrates the relative affect the mutation in each the $\Delta G$ of binding complex, $\Delta\Delta G_{binding} = \Delta G_{wild-type} - \Delta G_{alanine}$. The binding energy for each alanine mutant based on the MM-PBSA method is plotted.

bridges and further stabilizes with the Q42 side chain on the receptor (Fig 3C). A significant loss in the binding free energy is predicted as $\Delta\Delta G_{Q498A} = -13.64 \pm 2.53$ kcal mol$^{-1}$ by replacing these Q493 and Q498 positions with alanine, which make the Q498 key residue of a viral binding hotspot with respect to the less effective Q493, for which $\Delta\Delta G_{Q493A} = -8.77 \pm 2.88$ kcal mol$^{-1}$, reflecting the relevant roles played by these two residues at the binding interface.

Other important binding residues showing significant energy loss upon alanine modifications included the L455, F456, F486, T500, and Y505 positions (with a threshold of –2.5 kcal mol$^{-1}$). In these positions, the binding free energy losses caused by alanine substitutions can decrease in the energy range of 2.6 ~ 3.0 kcal mol$^{-1}$ (Fig 2). Furthermore, the side chain of the

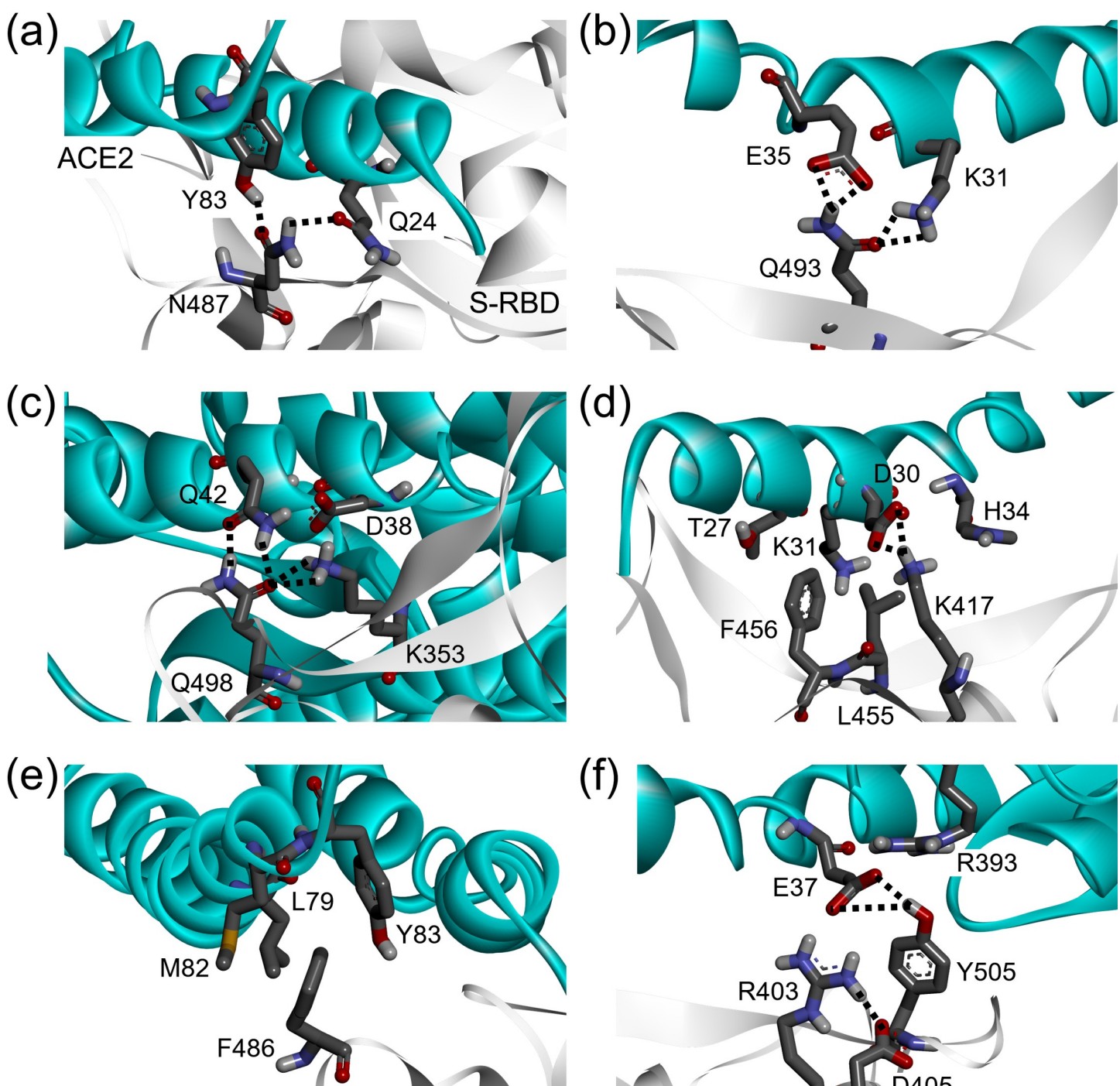

**Fig 3. Structural details of the protein-protein interaction involving the S-RBD and ACE2 residues at the binding interface obtained from the Prod phase MD simulations.** The secondary structures of S-RBD and ACE2 are portrayed as white ribbons and light-blue ribbons, respectively. Each interacting amino acid residue is highlighted in black-color labeled. The hydrogen bonds and salt bridges are represented in black and red dashed line, respectively.

L455 residue points to a charged pocket sealed with D30, K31, N33, and H34 of the ACE2 binding receptor through a moderate stabilizing vdW interaction, while F456 provides the intermolecular π-cation-stabilizing interaction with T27, D30, and K31 of ACE2 (Fig 3D) [65].

F486 presents to stabilize the hydrophobic side chain of Y83 on ACE2 binding, and also covers the neighboring side chain of L79 and M82 residues (Fig 3E). When all of these residues are replaced with alanine, a loss of the corresponding binding free energy is reflected as $\Delta\Delta G_{L455A}$ = −2.67 ± 2.14 kcal mol$^{-1}$, $\Delta\Delta G_{F456A}$ = −2.62 ± 1.29 kcal mol$^{-1}$, and $\Delta\Delta G_{F486A}$ = −2.95 ± 1.31 kcal mol$^{-1}$.

Similarly, Y505 interacts in the central segment of the binding interface with the side chain of the E37-R393 salt bridge on the receptor, while exchanging an internal π-cation interaction between an aromatic group of this residue and the guanidinium moiety of R403, making this interaction pattern reduce the binding affinity for the ACE2 receptor by ~3.0 kcal mol$^{-1}$ ($\Delta\Delta G_{Y505A}$ = −2.97 ± 2.52 kcal mol$^{-1}$) (Fig 3F). Besides, R403 provides faintly stabilizing polar or electrostatic interactions (occupancy 5%, distance$_{avg}$ = 3.228 Å) with the E37 side chain of ACE2 at the binding interface. The relative importance of these key residues agrees well with its interaction formation and clearly supports results on the interface structure and energetic framework in further experiments.

## Structural analysis and stability of the wild and mutant variants SARS-CoV-2 spike protein complexes

The crystal structure of the wild-type SARS-CoV-2 spike receptor-binding domain bound with ACE2 (PDB ID: 6M0J) was taken as the reference [40]. The structure of the wild-type complex along with the variants of the Kappa (L452R and E484Q), Delta (L452R and T478K), and Omicron (N440K, G446S, S477N, T478K, E484Q, Q493K, G496S, Q498R, N501Y, and Y505H) models were subjected to the MD simulation calculations. Various structural analyses were performed to study the structural impact of these variants. The final MD structures of each variant complex at the binding interface are illustrated in S1–S4 Figs.

Inspecting the dynamics and complex stability of the ACE2-bound S-RBD variant complexes, Root Mean Square Deviations (RMSD) with respect to their optimized structures were monitored for the simulated trajectories of all phase MD simulations, as shown in Fig 4A. Steady oscillations and small fluctuations of RMSD were observed in each complex model compared to the wild-type model. The RMSD plots for the wild-type complex were found to be the highest as compared with the values for the observed variants (S5 Fig). The wild-type model showed increases in the RMSD value at around 39 ns; however, after 45 ns, the RMSD gradually decreased and showed variable fluctuations until the end of the simulation. The slight conformational shift was seen in the Kappa and Omicron variants between 35–40 ns, which later stabilized [59]. Comparatively, Delta showed stable RMSD with inconsiderable fluctuations and endured lesser conformational changes during the whole MD simulations [24, 59, 60]. The average RMSD undulation amplitude was below 0.5 Å throughout most of the simulation time.

Furthermore, to explore the structural analysis of the potential model in sufficient detail, we calculated the dynamics of the residues in terms of the analysis of root mean square fluctuations (RMSF), which is useful to situate the flexible and disordered regions as well as the heterogeneity of the system [66, 67]. The plot showed a similar pattern of fluctuations with different magnitudes for all simulated systems (Fig 4B). The higher degree of evaluation was found in the native complex, while the Kappa variant witnessed the lowest level of fluctuations within the amino acid residues, indicating the better stability of the Kappa variant complex, followed by the Delta and Omicron variants [24, 59]. As the binding interface residues array in a random coil conformation lacks structural rigidity, a structural arrangement in this region should be necessary to sustain their configurations of the binding surface, which may facilitate the binding affinity. Three loop regions (residues 474–486, 488–490, and 494–505) of

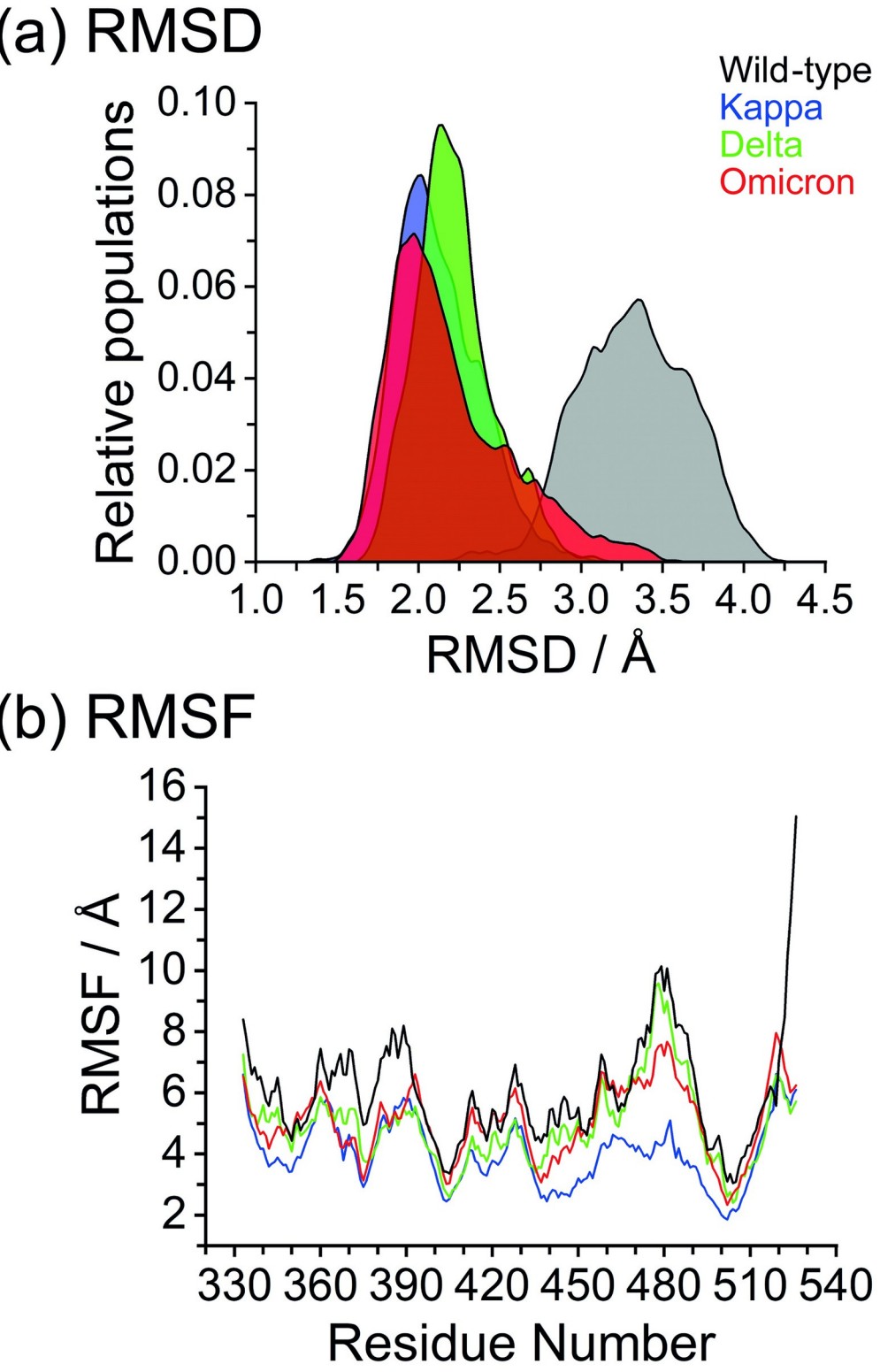

**Fig 4. The molecular dynamics results of the ACE2-bound S-RBD complex with the wild-type, Kappa, Delta, and Omicron variants over 50 ns simulation times.** (a) RMSD distribution, and (b) RMSF of Cα-atom of spike RBD; wild-type, Kappa, Delta, and Omicron, in complex with the ACE2. The positions of spike RBD-ACE2 binding region are denoted the fragment of residues 438–505.

SARS-CoV-2 demonstrated limited fluctuation compared to the corresponding region in the SARS-CoV wild-type [60]. Lower fluctuations, observed in the case of SARS-CoV-2 that bind to the K353 of ACE2, could be another reason for the better affinity of the S-RBD protein to ACE2 [40, 68]. The residues 480–488 (a random coil near the binding interface), showed are markable decrease in RMSF value for the variant models (S6 Fig). That is to say, these strong affinity positions stabilized the complex as a whole, indicating the reduced fluctuation in these interfacial residues for the ACE2 binding. In addition to terminal loops, the β-strand conformation (residues 510–524) displayed greater structural rigidity with less fluctuation than the wild-type, which mostly consisted of loops and might account for the binding affinity in each variant [58].

## Effect of SARS-CoV-2 variants on the hydrogen bond formation with ACE2

To evaluate the interfacial binding behavior on the ACE2-bound S-RBD variant complexes, the simulated trajectories were analyzed with the cut-off of 3.5 Å distances from acceptor to donor atomic pairs that play critical roles in the stronger binding tendency towards the ACE2 receptor compared to the wild-type. Here, three mutagenic viruses in the S-RBD protein of SARS-CoV-2, currently designated as VOC by WHO, *i.e.*, B.1.617.1 (Kappa), B.1.617.2 (Delta) and B.1.1.529 (Omicron), were simulated. The evolution and stability of the hydrogen bond formations between the binding interface residues were monitored as a function of MD simulation trajectory to trace the occupancy and hydrogen bond formation, in which more than a 10% occupancy rate was shown, as listed in Table 2. Recently, the relative strength and hydrogen bond variation between each atomic pair have been evaluated computationally with the findings substantially in line with our measurements [21, 24, 59, 63, 69]. That is to say, the high-affinity protein-protein complex of S-RBD toward ACE2 is attributed to the explanation of why the binding affinities are distinctly different from each other.

Based on the analysis of hydrogen bonding at the S-RBD binding interface with ACE2 (cut-off 3.5 Å), we observed that the mutant residue was found to have a cascading effect on the interfacial surface of the S-RBD region. A similar pattern of an amino acid pairing on the S-RBD regions, *i.e.*, Y449, N487, Q493, Q498, T500, and Y505, was monitored as key hotspots for all complex systems (bold in Table 2), which are oriented in the potential site through negligible hydrogen bonding with two-salt bridges at the ACE2 binding [63, 64, 70–72]. In addition, the high average number of hydrogen bonds at the interfacial binding surface of the Kappa and Delta variants can maintain the stronger interaction of the ACE2-bound S-RBD complex, which is relatively higher compared with the wild-type [24]. Apart from these residues, the Kappa variant has two additional residues (S477 and T478), the Delta variant has five additional residues (Y473, Y489, F490, Y495, and N501), and the Omicron variant showed more two additional residues (Y453 and S496), that were found to be involved in forming the intermolecular hydrogen-bonded interactions [24, 73]. Moreover, the Q493K mutation in the Omicron RBD protein resulted in the loss of the salt bridge with K31 and the hydrogen bond with Q42 of ACE2 [59], while the contact with D38 was newly formed. The likelihood of an interaction between the Q493 of SRAS-CoV-2 and K31 residues of ACE2 was reported recently [60, 68]. In the case of the G446S mutant, no interaction was observed during the simulation as supported by the previous finding. The Q493 position in all complex systems interacts with E35 of the ACE2 receptor with significant occupancy. T500 of the wild-type, Kappa, and Omicron variants mediates a hydrogen bond with D355 of the ACE2 with high occupancy, while the interaction between T500 of Delta and Y41 of ACE2 was observed. Our finding was consistent with previous studies [59, 74, 75]. However, some contradictorty

**Table 2. Hydrogen bond occupancy for the residue pairs at the S-RBD binding interface with ACE2 (cut-off 3.5 Å) over the production phase MD simulations.**

| Atomic pairs | HB Occupancy / % | Average Distance$_{A-D}$ / Å |
|---|---|---|
| **Wild-type** | | |
| **[ACE2] GLN42@OE1: [RBD] GLN498@HE21** | 92.56 | 3.011 |
| [RBD] GLN498@OE1: [ACE2] GLN42@HE21 | 85.45 | 2.881 |
| **[RBD] ASN487@OD1: [ACE2] TYR83@HH** | 83.7 | 2.792 |
| **[ACE2] ASP38@OD1: [RBD] TYR449@HH** | 78.94 | 2.727 |
| **[ACE2] ASP355@OD2: [RBD] THR500@HG1** | 76.92 | 2.811 |
| **[ACE2] GLU37@OE2: [RBD] TYR505@HH** | 54.96 | 2.794 |
| **[ACE2] GLU35@OE2: [RBD] GLN493@HE21** | 44.73 | 2.922 |
| **[RBD] GLN498@OE1: [ACE2] LYS353@HZ1** | 40.68 | 2.826 |
| [ACE2] GLU35@OE1: [RBD] GLN493@HE21 | 37.81 | 2.926 |
| [ACE2] HIE34@ND1: [RBD] TYR453@HH | 36.13 | 2.874 |
| [ACE2] ASP38@OD2: [RBD] TYR449@HH | 34.23 | 2.902 |
| [RBD] GLN498@OE1: [ACE2] LYS353@HZ3 | 34.12 | 2.820 |
| **[ACE2] GLN24@OE1: [RBD] ASN487@HD21** | 27.98 | 2.962 |
| [ACE2] GLU37@OE1: [RBD] TYR505@HH | 26.54 | 2.894 |
| **[RBD] GLN493@OE1: [ACE2] LYS31@HZ2** | 0.1497 | 14.97 |
| [RBD] GLN493@OE1: [ACE2] LYS31@HZ3 | 0.1327 | 13.27 |
| **Kappa (B.1.617.1)** | | |
| **[RBD] ASN487@OD1: [ACE2] TYR83@HH** | 94.21 | 2.786 |
| **[ACE2] ASP355@OD2: [RBD] THR500@HG1** | 61.31 | 2.770 |
| [RBD] GLN498@OE1: [ACE2] GLN42@HE21 | 56.94 | 2.891 |
| **[ACE2] GLN42@OE1: [RBD] GLN498@HE21** | 49.62 | 3.017 |
| **[ACE2] GLU35@OE2: [RBD] GLN493@HE21** | 48.9 | 2.906 |
| **[ACE2] GLU35@OE1: [RBD] GLN493@HE21** | 39.59 | 2.915 |
| **[ACE2] ASP38@OD2: [RBD] TYR449@HH** | 37.34 | 2.767 |
| **[ACE2] GLU37@OE1: [RBD] TYR505@HH** | 36.63 | 2.793 |
| [ACE2] ASP38@OD1: [RBD] TYR449@HH | 36.19 | 2.801 |
| [ACE2] TYR41@OH: [RBD] THR500@HG1 | 34.76 | 2.838 |
| [ACE2] GLN24@OE1: [RBD] SER477@H | 34.33 | 2.973 |
| **[RBD] GLN498@OE1: [ACE2] LYS353@HZ1** | 22.03 | 2.847 |
| **[RBD] GLN493@OE1: [ACE2] LYS31@HZ1** | 14.58 | 2.829 |
| **[RBD] GLN493@OE1: [ACE2] LYS31@HZ2** | 13.49 | 2.831 |
| **[RBD] GLN498@OE1: [ACE2] LYS353@HZ2** | 11.63 | 2.849 |
| [ACE2] HIE34@ND1: [RBD] GLN493@HE22 | 10.86 | 2.986 |
| [RBD] THR478@OG1: [ACE2] GLN24@HE22 | 10.19 | 2.988 |
| **Delta (B.1.617.2)** | | |
| [ACE2] TYR83@OH: [RBD] TYR489@HH | 97.61 | 2.841 |
| **[RBD] ASN487@OD1: [ACE2] TYR83@HH** | 96.15 | 2.732 |
| [ACE2] TYR41@OH: [RBD] THR500@HG1 | 59.94 | 2.821 |
| **[ACE2] GLU35@OE2: [RBD] GLN493@HE21** | 51.81 | 2.873 |
| [ACE2] ASP38@OD1: [RBD] GLN498@HE21 | 49.08 | 2.875 |
| [ACE2] ASP38@OD2: [RBD] GLN498@HE21 | 48.22 | 2.887 |
| **[ACE2] ASP355@OD2: [RBD] THR500@HG1** | 40.96 | 2.759 |
| **[ACE2] GLN24@OE1: [RBD] ASN487@HD21** | 38.3 | 2.963 |
| **[ACE2] GLU37@OE1: [RBD] TYR505@HH** | 35.21 | 2.870 |
| [ACE2] GLU23@OE1: [RBD] TYR473@HH | 34.58 | 2.741 |

*(Continued)*

**Table 2.** (Continued)

| Atomic pairs | HB Occupancy / % | Average Distance$_{A-D}$ / Å |
|---|---|---|
| [ACE2] GLU37@OE2: [RBD] TYR505@HH | 34.05 | 2.903 |
| **[ACE2] GLU35@OE1: [RBD] GLN493@HE21** | 31.87 | 2.884 |
| **[RBD] GLN498@OE1: [ACE2] LYS353@HZ1** | 30.76 | 2.879 |
| **[RBD] GLN498@OE1: [ACE2] LYS353@HZ2** | 29.21 | 2.877 |
| [RBD] TYR495@O: [ACE2] LYS353@HZ1 | 26.14 | 3.000 |
| [ACE2] GLU23@OE2: [RBD] TYR473@HH | 22 | 2.754 |
| [RBD] TYR473@OH: [ACE2] SER19@HG | 21.79 | 2.919 |
| [ACE2] TYR41@OH: [RBD] ASN501@HD22 | 19.88 | 3.259 |
| [ACE2] HIE34@O: [RBD] GLN493@HE22 | 18.9 | 2.978 |
| [RBD] PHE490@O: [ACE2] LYS31@HZ1 | 12.97 | 2.898 |
| **Omicron (B.1.1.529)** | | |
| **[ACE2] ASP355@OD2: [RBD] THR500@HG1** | 76.93 | 2.773 |
| [ACE2] HIE34@ND1: [RBD] TYR453@HH | 58.97 | 2.912 |
| **[RBD] ASN487@OD1: [ACE2] TYR83@HH** | 49.93 | 2.801 |
| [ACE2] ASP38@OD1: [RBD] SER496@HG | 43.71 | 2.658 |
| **[ACE2] GLU35@OE2: [RBD] LYS493@HZ2** | 32.14 | 2.813 |
| **[ACE2] GLU35@OE2: [RBD] LYS493@HZ3** | 29.99 | 2.792 |
| [ACE2] ASP38@OD2: [RBD] LYS493@HZ1 | 24.06 | 2.782 |
| [ACE2] ASP38@OD1: [RBD] LYS493@HZ3 | 23.26 | 2.786 |
| [ACE2] TYR41@OH: [RBD] THR500@HG1 | 21.05 | 2.897 |
| [ACE2] GLN24@OE1: [RBD] ASN487@HD21 | 17.43 | 2.984 |
| [ACE2] ASP38@OD2: [RBD] LYS493@HZ2 | 16.17 | 2.796 |
| [ACE2] ASP38@OD2: [RBD] LYS493@HZ3 | 15.21 | 2.799 |
| [RBD] ASN487@OD1: [ACE2] GLN24@HE22 | 14.53 | 3.010 |
| [ACE2] GLU35@OE2: [RBD] LYS493@HZ1 | 13.16 | 2.790 |

Average distance$_{A-D}$ = average distance between acceptor and donor atom.

Bold atomic pairs represent the similar hydrogen bond interaction in each variant related to the wild-type.

observations were also found. Notably, residues L455, F456, F486, and S494, which were reported to form hydrogen bonds and electrostatic interactions leading to an enhanced binding affinity of SARS-CoV-2, were not observed in these simulations [39, 56, 68].

The structural environment changes in the Q493 beta-sheet can be observed according to the L452R mutation in the Delta variant (Fig 5A) [73]. We found that the number of hydrogen bond occupancies of Q493 with ACE2 at the binding interface increased significantly for the variants that express the L452R mutation (Fig 5B), while the Kappa variant changed slightly. Although the Omicron variant did not mutate at the L452, we found an increased amount of intermolecular hydrogen bonding of Q493 for ACE2 binding, which might be due to this variant carrying the Q493K mutation. That is to say, the changing of the structural environment neighbor 493 position and/or by itself can enhance binding affinity through hydrogen bond formation with the ACE2 protein relative to the Delta. The basic side chain of mutated R452 orients itself for stabilizing and increases the hydrogen bond interaction with the salt-bridge residues (K31 and E35) by Q493 [76, 77]. The oxygen atom of the Q493 side chain creates hydrogen-bonding interaction with the K31 side chain, while one of the hydrogens of the amine group forms the intermolecular hydrogen bonds with the carboxyl group of E35 from ACE2 (Fig 5C). In addition, we also mentioned that the Q493K mutation in the Omicron RBD protein presented several intermolecular hydrogen bonds with E35, and the contact with D38

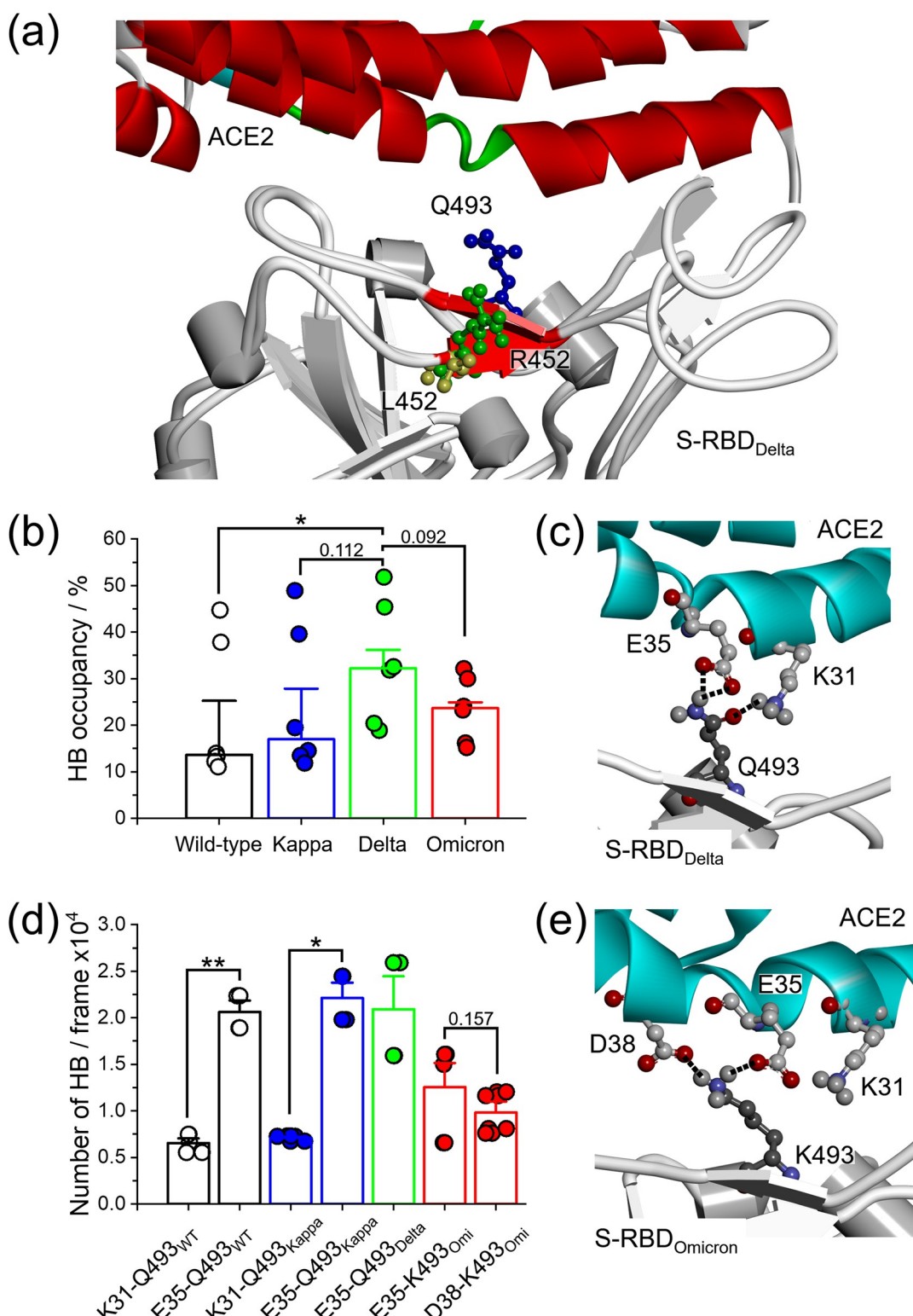

**Fig 5. Hydrogen bonding evolution in the Delta and Omicron variants.** (a) Superimposition between S-RBD of the Delta and the wild-type that represents the Q493 (blue), L452 in the wild-type (yellow), and R452 in the Delta (green) as expressed on two neighboring β-strands forming a small antiparallel β-sheet (red). (b) Combined analysis of the wild-type and three variants that carry on the L452R exchange with regarding to hydrogen bond occupancies (%) formed by Q493 with ACE2 ($^*$ $p < 0.05$, two-tailed Student's T-test). (c) Hydrogen bond formation represents between Q493 in S-RBD of the Delta and K31

and E35 salt bridges at binding interface with ACE2. (d) The number of the intermolecular hydrogen bond between Q493 (or K493 in the Omicron) from S-RBD and ACE2 at K31, E35, and D38 (* $p < 0.05$, ** $p < 0.005$, two-tailed Student's T-test). (e) Structural representation of the two newly formed salt bridges between K493 on the S-RBD in the Omicron and E35 and D38 that expressed on ACE2. The hydrogen bonds are indicated by black dashed line.

was newly formed, which was not observed in the wild-type residue expressing a glutamine 493 (Fig 5D and 5E, and S7 Fig). However, the stable salt-bridge with K31 was not observed, which might be too short of a lysine side chain for ACE2 binding [58]. Thus, the Omicron variant can form stable salt-bridges between Q493 and E35 and D38 of ACE2 interaction.

## Energetic affinity of the binding interfaces with ACE2

Supporting the previously stated results, we also calculated the values of binding free energy ($\Delta G_{binding}$, kcal mol$^{-1}$) obtained from the MM-PBSA method to compare the binding interface behavior of the mutated S-RBD protein relative to the wild-type [24, 58, 64, 70–74, 78]. The estimated binding free energies ($\Delta G_{binding}$) calculated for the ACE2-bound S-RBD complex for each variant are listed in Table 3. It was interesting to note that the heavily mutated Delta protein (Delta, Omicron, and Kappa variants) showed the strongest binding affinity of –73.46 kcal mol$^{-1}$ followed by the Omicron (–61.75 kcal mol$^{-1}$) and Kappa (–58.90 kcal mol$^{-1}$) variants. Although, our results are a little different from the previous literature that proposed the highest binding affinity for the Omicron variant [59], the $\Delta G_{binding}$ magnitude of the mutated S-RBD proteins was also found to be lower than that of the wild-type complex by –5.79 - – 20.35 kcal mol$^{-1}$. Furthermore, the calculated $\Delta G_{binding}$ value is in a good agreement with the findings of previous work that showed greater cell-surface binding affinity between the mutated S-RBD (N510Y, K417N, and E484K) and the ACE2 protein than the wild-type by a competition binding assay [73]. The enhanced interaction of the Omicron S-RBD protein with ACE2 is consistent with previously published data [79], and may contribute to the increased infectivity of the Omicron variant. Our result again emphasized the role of the multimutated S-RBD protein in increasing binding affinity toward the ACE2 receptor as supported by the findings in previous studies [58, 73, 74, 78].

Furthermore, the individual energy term (vdW, polar, and nonpolar solvation free energies) was also calculated from the total binding free energy to understand the contribution of individual energy components in the binding. The results revealed that the main driving force for increasing the binding affinity was compensated by favorable change in the electrostatic contributions (EEL) and van der Waals (vdW) interactions. This energy term was found to relate to the analysis of the electrostatic interaction energy at the 493 key residue that showed a strong increase for Omicron when compared to others (S8 Fig), supporting the structural

**Table 3. Free energy terms (kcal mol$^{-1}$) for SARS-CoV-2 variants on the S-RBD protein binding to ACE2 estimated by MM-PBSA method.**

| Parameters | Kappa | Delta | Omicron | Wild-type |
|---|---|---|---|---|
| $\Delta G_{binding}$ | −58.90±7.313 | −73.46±8.222 | −61.75±6.679 | −53.11±6.701 |
| vdW | −94.35±4.698 | −109.81±5.184 | −92.59±5.763 | −86.89±5.911 |
| EEL | −1067.10±32.451 | −1081.70±28.234 | −1482.13±36.602 | −714.37±21.473 |
| EPS | 1113.34±30.341 | 1129.14±24.535 | 1528.44±36.775 | 758.37±20.712 |
| ENPOLAR | −10.78±0.326 | −11.09±0.292 | −10.47±0.388 | −10.21±0.350 |

**Note:** The vdW and EEL represent van der Waals and the electrostatic contributions from MM, respectively. EPS stands for PB electrostatic contribution to the polar solvation free energy, while ENPOLAR is the nonpolar contribution to the solvation free energy. $\Delta G_{binding}$ (kcal mol$^{-1}$) is the final estimated binding free energy calculated from the terms above.

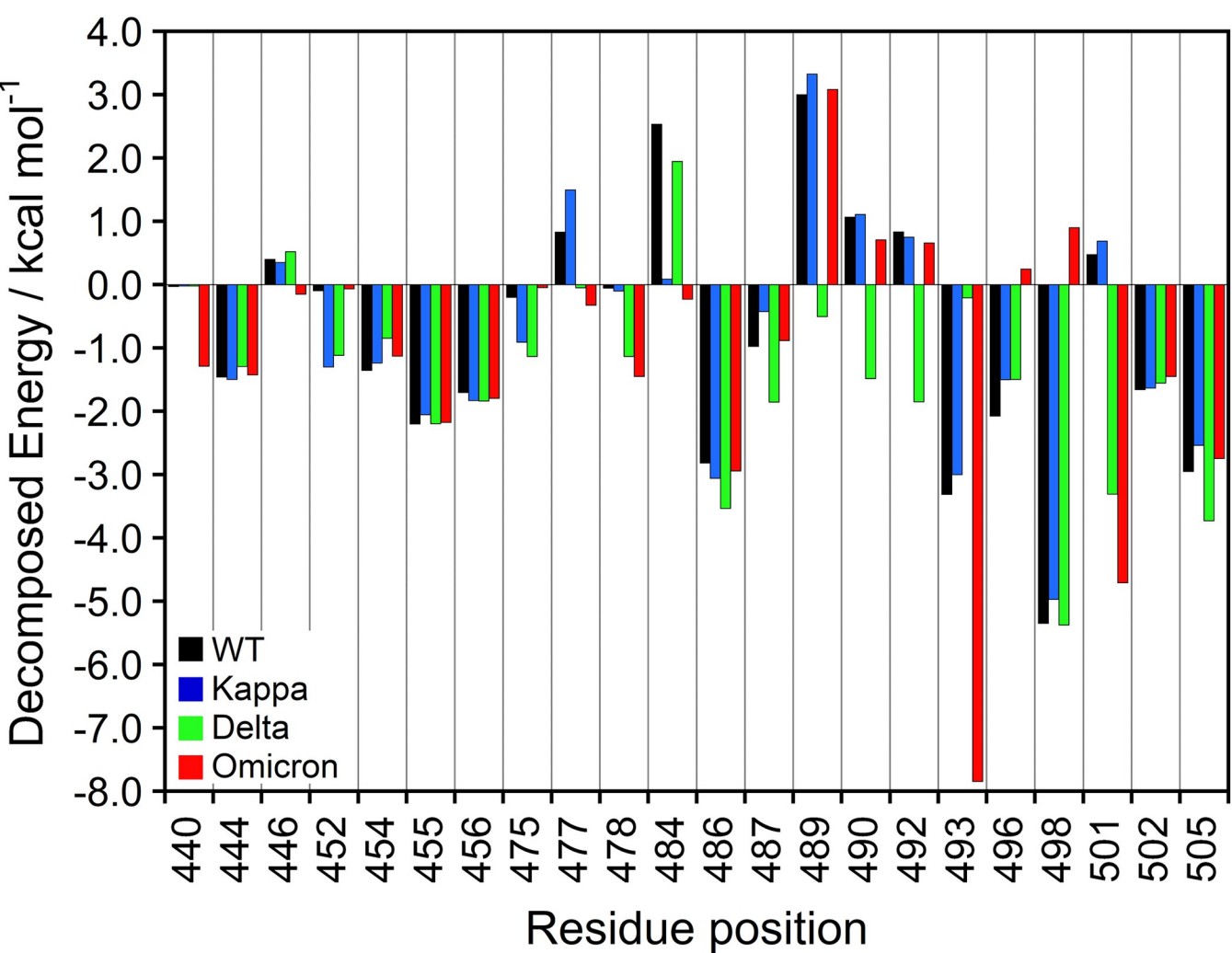

**Fig 6. Per-residue free energy decomposition of the key hotspot at the S-RBD domain (residues 438–505) of the binding interface with ACE2 protein.** All values were given in kcal mol$^{-1}$.

environment changes of Q493 as the result of L452R mutation. The sum of EEL and the polar component of solvation (EPS) contributions is more positive for the mutations, indicating that stabilization is mostly due to the hydrophobic interactions. Taken as a whole, the structural analysis could show the binding characteristics for the ACE2-bound S-RBD variant interaction compared to the wild-type SARS-CoV-2. Specifically, the high mutation rate in the Omicron S-RBD corresponding with the electrically charged side-chains leads to an improved binding affinity for the ACE2 interaction compared to the Delta variant, which may allow the Omicron variant to facilitate higher transmissibility in contrast to the wild-type.

With the help of the structural information provided by the analysis of the binding affinity, the binding free energies were decomposed into the energy contributions of the individual residues at the binding interface, which hopefully will allow researchers to detail the cross-reactivity of the neutralizing antibody response. Fig 6 illustrates the decomposed per-residue free energy (kcal mol$^{-1}$) calculated by the MM-PBSA method upon the interfacial residues of the S-RBD regions (at position 438–505) with ACE2 receptor (S8 Fig). The negative and positive energy values represent favorable and unfavorable contributions, respectively.

In addressing the highest mutagenic rate in the Omicron variant, the dominant mutational Q493K and N501Y hotspots highlighted more favorable interactions with strong binding affinity for the ACE2 interaction. Consequently, we believe that N501Y is a critical mutation that affects the transmission of SARS-CoV-2 by strengthening the interaction between S-RBD and the ACE2 protein [80]. The strong interaction of the N501Y mutant leads to the tighter binding of SARS-CoV-2 to the host cell, allowing complete membrane fusion or the internalization of the receptor with the virus. Several other S-RBD variants, *i.e.*, Kappa and Delta that have very high transmission rates even though they do not contain the N501Y mutation, might increase the binding affinity for the ACE2 receptor, leading to similar effects [81]. In addition, the other mutated residues (N440K, S477N, T478K, and Y505H) were found to significantly promote favorable interaction to the binding of the Omicron S-RBD complex with ACE2 protein [59], while the mutated G446S and E484Q positions contributed less to the interaction increment than those of the mutated residues in the Omicron variant, which is related to the previous experimental and theoretical analysis [73]. On the contrary, the decomposed energy forms unfavorable contact with ACE2 binding even though the mutual residues of G496S and Q498R slightly shift position. This interesting result implied the impact of these mutagenic positions for higher infectivity and transmission of the Omicron variant.

Similarly, L452 and T478, which are mutated to L452R in Kappa, and T478K in the Delta variant, showed high potential individual decomposed energy by mediating a consistent hydrogen bond [59]. R454, L455, and F456 showed significantly less contribution difference in the binding energy for all systems. The Delta mutant also specifically promoted the dominant positions, N487, Y489, F490, and L492, with stronger interaction at the binding interface of the complex compared to the others. Although these positions together with F486, N487, and Y505 presented the lowest binding free energy values in agreement with the hydrogen bonding analysis (Table 2), they did not present mutagenicity in the Delta and Kappa variants.

Previous studies have proposed that different neutralizing antibodies target different regions on the S-RBD protein [24, 82]. The epitome mapping analysis on the monoclonal antibody specific to the spike protein revealed that 20% of more than 400 epitopes were derived from the RBD region. The mutated S-RBD protein in the Kappa, Delta, and Omicron variants of SARS-CoV-2, which contains the ACE2 binding site as well as the epitopes over 80% of antibodies induced by infection or vaccination, decrease the binding affinity of the antibody [23]. For instance, the L452R mutation in the Delta and Kappa variants was found to be the center position of the JMB2002-binding epitope with Y102 from the heavy chain of the Fab, thus providing an explanation for its loss of potency against the mutation [25]. It should be noted that the structural inspection alone may not be sufficient for identifying the key contributions to binding affinity, where the effects of the solvation term are considered. These coevolving residues can mediate protein recognition in multiprotein complexes and are often spatially close to each other, forming clusters of interacting residues that are located near functionally important sites [83, 84].

## Dynamic cross-correlation matrix and conformational flexibility analysis

To further investigate the conformational dynamics in detail, the dynamic cross-correlation matrix (DCCM) was calculated to measure the occurrence of correlated motions according to the simulation period based on the positions of backbone carbon–atoms of the S-RBD protein. The phenomenon of the dynamic motions during the simulations of the Cα–atoms of the SARS-CoV-2 spike RBD protein in each bound system are shown in Fig 7. Highly correlated motions, also mentioned as positive correlation, range from the color green (low), through yellow (medium), to deep red (high, +1), while anti-correlated motions, also represented as

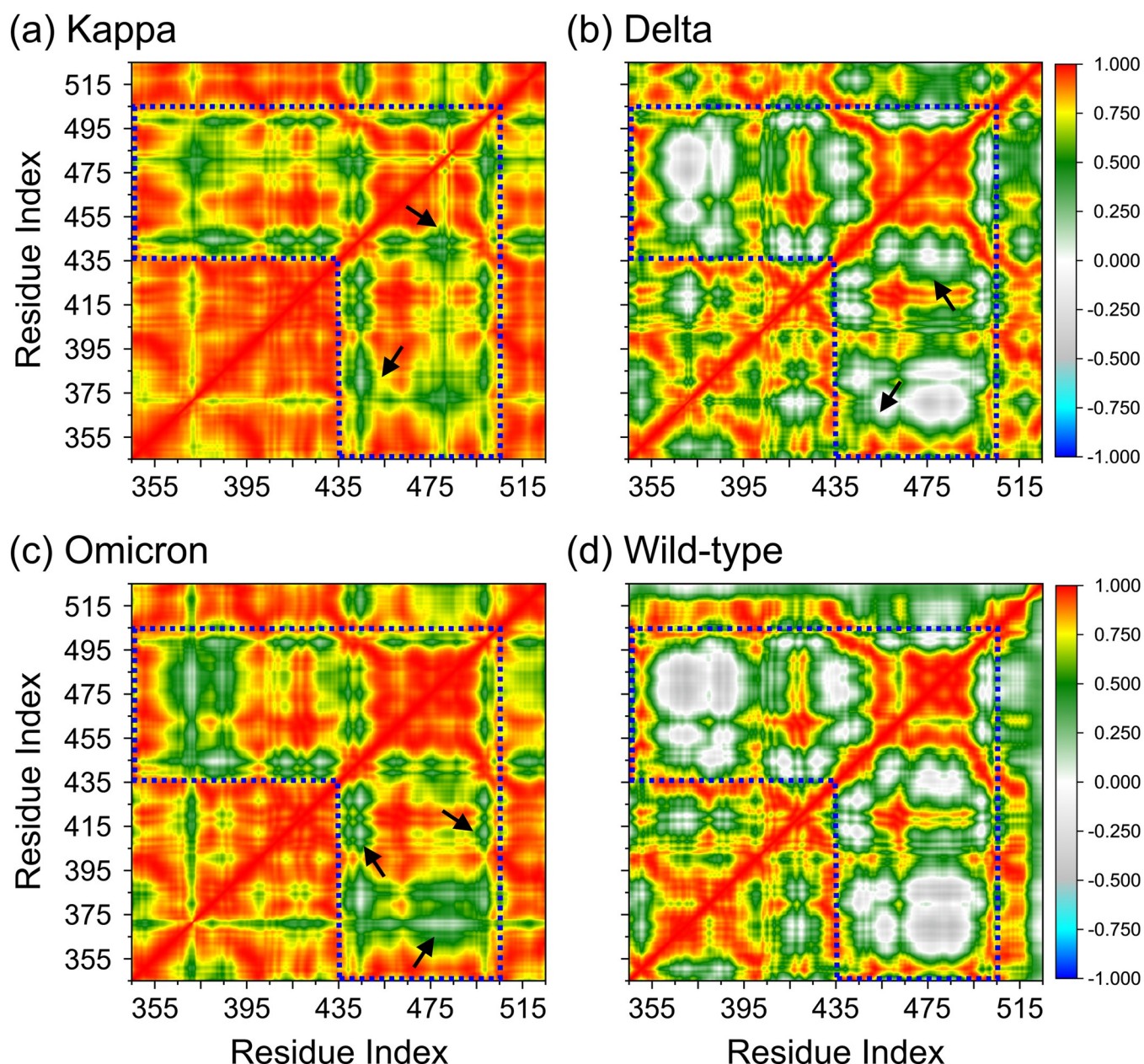

**Fig 7.** Dynamic cross-correlation diagrams of the fluctuations of Cα–atoms of SARS-CoV-2 spike RBD protein in the various variants of (a) Kappa, (b) Delta, (c) Omicron in comparison to the (d) wild-type RBD protein. Positive and negative values are represented in range of color deep red to royal blue, respectively. The diagonal square relates to the correlation of a residue with itself, *i.e.*, only a region remarked to have highly-positive values (red). Black arrows point to the mutation position of each variant that observed for ACE2 binding. The ACE2 binding regions of the SARS-CoV-2 spike RBD protein, at residues number of 438–505, are demarcated with dashed lines.

negative correlation, range from gray (low), through cyan (medium), to royal blue (high, –1). The diagonal square corresponds to the relationship of a residue with itself, *i.e.*, the only region observed to present highly positive values (deep red).

The DCCM map results revealed that the S-RBD variants can affect the structural remodeling for the ACE2-bound S-RBD complex, as illustrated by the change in the dynamic patterns and correlations [51, 85]. As shown in dashed lines in Fig 7, three S-RBD variants dominated

primarily by correlated motion (lesser gray region) in comparison to the wild-type system. Overall, the Kappa and Omicron conformations display strong correlated motions (red and yellow), while the Delta conformations are embodied by low correlated motion (green). The black arrows point to the mutation position of each variant that was observed for ACE2 binding. The biggest differences observed from the correlated motion maps are that the Omicron variant triggered a correlated dynamic on the binding site as opposed to the Delta and Kappa variants. Since the terminal loops at residues 510–524 accounted for the ACE2 binding based on the structural rigidity (Fig 4B), a more widely correlated motion is observed in the S-RBD variant protein. The above regions relating with obvious changes of motion mode agree with the RMSF changes of the S-RBD variant proteins. These changes of the internal dynamics may reflect different alternation of relative positions between key residues, which may play an important role in the conformational diversity of the S-RBD variants. Furthermore, this finding also explains the lower decomposition binding free energy for the binding site observed during the MM-PBSA calculation and how they were compensated by the additional contacts from the mutation region.

Considering the fact that the biological function of a protein-protein interaction is influenced by its conformational dynamics, we also employed Principal Component Analysis (PCA) as the computational method to investigate the conformational transitions of each complex system using the diagonalization of the covariance matrix of the Cα–atoms fluctuations over the Prod-phase MD simulation. Fig 8 shows the PCA scatter plot generated for the ACE2-bound SARS-CoV-2 spike RBD wild-type and their spike RBD variant complexes along the first two principal components (PC). Most of the simulation ensembles in the variant complex related with the strong binding affinity are concentrated to a narrow range of the conformational space compared to the wild-type [86–88]. This suggests that the variant complex offers more stability and compact packing than the wild-type model, which corroborates the previous reports on the compact cluster of stable states in the variants [24]. Overall, the less correlated motion of the wild-type system confers with the observed higher residue flexibility in Fig 6D, implying that the mutational behaviors of the S-RBD variants triggered conformational dynamics as conferred by the conformational flexibility and dynamic cross-correlation analysis. The variation in structural motions allowed us to assess the binding phenomena of each S-RBD variant for ACE2 binding.

## Conclusions

In the present work, the S-RBD variants in both Omicron and Delta were investigated by focusing on the effectiveness of the structural analysis using several computational tools and a computational saturation mutagenesis structure, as well as the dynamic changes that are likely to contribute to both the protein stabilization and binding affinity of the S-RBD protein for ACE2 interaction. Based on the analysis of the amino acid composition, the amino acids with electrically charged side chains (arginine (R), lysine (K), histidine (H), and glutamic acid (E)) were increased in Omicron, which may increase the contribution of the salt bridge of ACE2 at the binding interface. Consequently, we performed systematic alanine scanning on all different residues in the S-RBD region of the wild-type SARS-CoV-2 structures that form most of the protein binding interfaces, and evaluated the variation in the corresponding binding free energy. These mutagenesis studies provide a clearer picture of the conserved molecular hotspots at the binding interface in the S-RBD protein of SARS-CoV-2 (wild-type) and highlight residues Q498, Q493, and N487 on the spike protein of the SARS-CoV-2 receptor-binding domain as potential residues contributing to the shaping and determination of stability in the relevant protein-protein binding interface.

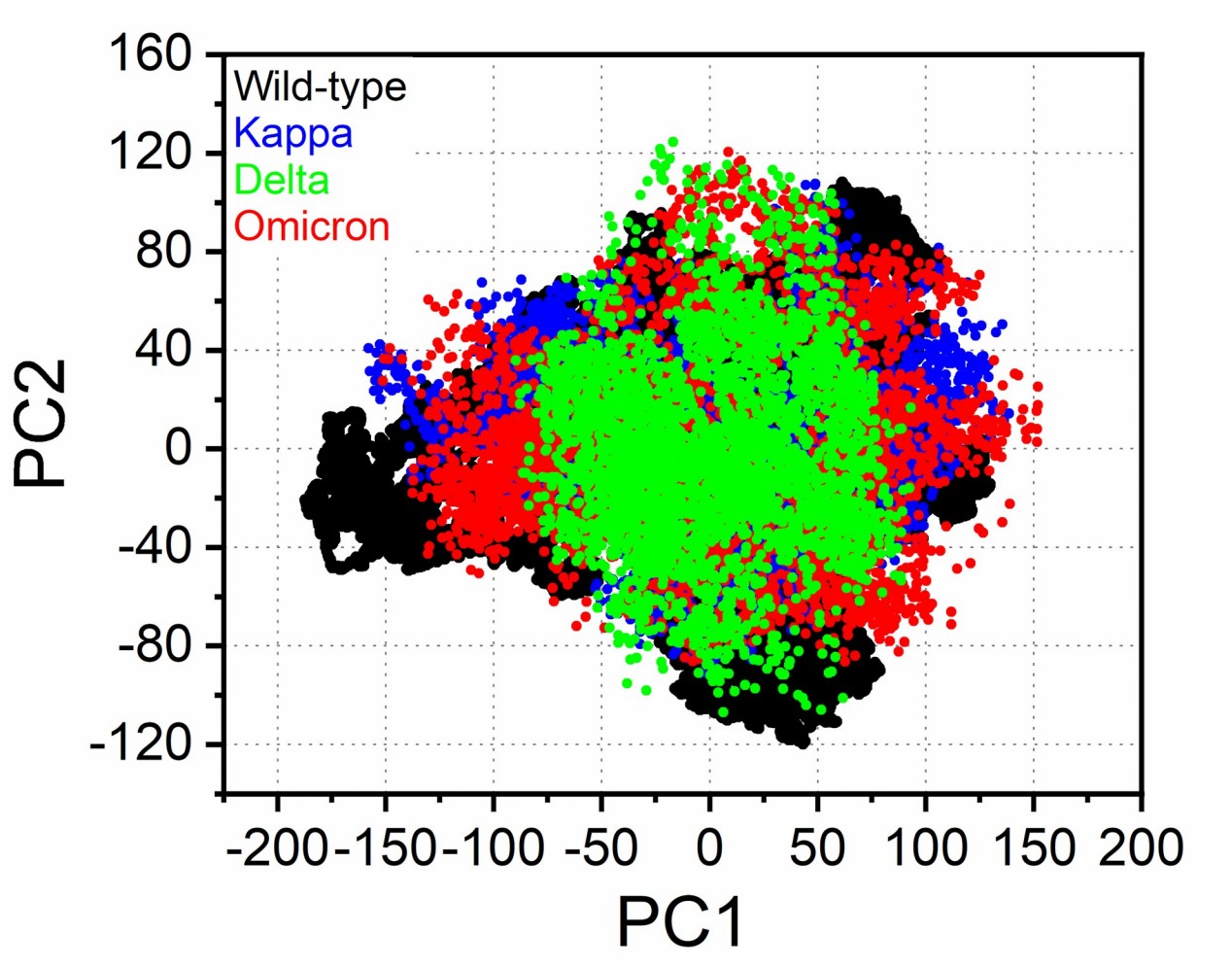

**Fig 8. PCA projection of the motion of Cα–atoms of the RBD-bound S-RBD complex.** The PCA profiles were obtained by plotting the first two principal components (PC1 and PC2) in the various conformation of the Kappa (blue), Delta (green), Omicron (red) and the wild-type (black) systems. PC1 and PC2, respectively, represent a covariance matrix after elimination of eigenvectors. Each point between the single-directional motions represents a unique conformation during the simulation, therefore, similar structural conformations overlap in the graph.

Another challenge in this study was providing structural-based information and an energetic framework at the binding interface of the spike-ACE2 complexes. A detailed analysis of hydrogen bonding formation, the estimated binding affinity of the binding complex, and the per-residues decomposition free energies indicated that the S-RBD variants (Delta and Omicron) interacted closely with the ACE2 protein at the binding interface. Nowadays, most vaccines are designed on the basis of the S-RBD protein, leading to the application of the neutralizing antibodies for targeting the S-RBD to weaken its ACE2 binding. We found that the potential hotspots in the S-RBD mutation had the strongest binding affinity compared to the wild-type for ACE2. Although the Omicron S-RBD disrupted some favourable hotspot residues for ACE2 binding, such variants can promote new advisable binding interactions. Additionally, some hydrogen bond formation and π-stacking interactions are formed, which can further encourage the S-RBD protein for ACE2 binding. Additional mutations within the

receptor-binding site (RBD region) may affect changes in the amino acid sequence of the epitope and may contribute to the escape from the neutralizing antibody binding. Optimistically, the information and results provided by this work will lead to a concrete informative article for viral community scientists occupied in the context of structure-based design and therapeutic neutralizing antibodies as well as vaccine development, since this information is urgently required in the current fight against future pandemics.

## Supporting information

**S1 Fig. Structural mapping at the binding interface structure of the wild-type S-RBD protein complex with ACE2.** The structure of ACE2 is shown in gray ribbon, while S-RBD is in green ribbon. The close-up of the binding interfaces is shown the crucial hotspots responsible for binding interaction (right panel). The interfacial residues of SARS-CoV-2 S-RBD residues are annotated and shown in green sticks, while ACE2 residues are in gray sticks. The figure was drawn by Discovery Studio 2019 Client (Biovia software).
(TIF)

**S2 Fig. Structural mapping at the binding interface structure of the Kappa S-RBD protein complex with ACE2.** The structure of ACE2 is shown in gray ribbon, while S-RBD is in green ribbon. The close-up of the binding interfaces is shown the crucial hotspots responsible for binding interaction (right panel). The interfacial residues of SARS-CoV-2 S-RBD residues are annotated and shown in green sticks, while ACE2 residues are in gray sticks. The figure was drawn by Discovery Studio 2019 Client (Biovia software).
(TIF)

**S3 Fig. Structural mapping at the binding interface structure of the Delta S-RBD protein complex with ACE2.** The structure of ACE2 is shown in gray ribbon, while S-RBD is in green ribbon. The close-up of the binding interfaces is shown the crucial hotspots responsible for binding interaction (right panel). The interfacial residues of SARS-CoV-2 S-RBD residues are annotated and shown in green sticks, while ACE2 residues are in gray sticks. The figure was drawn by Discovery Studio 2019 Client (Biovia software).
(TIF)

**S4 Fig. Structural mapping at the binding interface structure of the Omicron S-RBD protein complex with ACE2.** The structure of ACE2 is shown in gray ribbon, while S-RBD is in green ribbon. The close-up of the binding interfaces is shown the crucial hotspots responsible for binding interaction (right panel). The interfacial residues of SARS-CoV-2 S-RBD residues are annotated and shown in green sticks, while ACE2 residues are in gray sticks. The figure was drawn by Discovery Studio 2019 Client (Biovia software).
(TIF)

**S5 Fig. RMSD profiles of MD simulations.** RMSD plot of each S-RBD variant complex with ACE2 protein over the production (Prod) phase MD simulations.
(TIF)

**S6 Fig. Superimposed structure of wild-type and mutated S-RBD protein.** The superimposition structure of the Cα-atom trace of four different structures between the wild-type and mutated S-RBD protein of SARS-CoV-2. The proteins superimpose almost exactly in most parts of the protein expect few regions which exhibit conformation variability (highlighted in the close-up), especially, at the RBD-ACE2 binding interfaces.
(TIF)

**S7 Fig. Hydrogen bonding evolution in the Delta and Omicron variants.** The intermolecular hydrogen bond occupancy between Q493 (or K493 in the Omicron) from S-RBD protein and ACE2 at K31, E35, and D38 for binding complex ($^*$ $p < 0.05$, $^{**}$ $p < 0.005$, two-tailed Student's T-test).
(TIF)

**S8 Fig. Energy decomposition analysis of wild-type and mutated S-RBD protein in complex with ACE2.** Per-residue decomposed energy of (a) electrostatic, and (b) vdW interaction on the key hotspots of the S-RBD regions (at position 438–505) for ACE2 binding. All values were given in kcal mol$^{-1}$.
(TIF)

**S1 Table. Computational alanine scanning mutation analysis.** Relative binding free energy terms (kcal mol$^{-1}$) calculated by the computational alanine scanning mutagenesis using MM-PBSA method approach for the S-RBD regions (at position 438–505) of SARS-CoV-2 residues effectively involved in the binding interface with the ACE2.
(DOCX)

## Acknowledgments

Authors would like to acknowledge the Erawan HPC Project, Information Technology Service Center (ITSC), Chiang Mai University, and Thailand Excellence Center for Tissue Engineering and Stem Cells (PK), Department of Biochemistry, Faculty of Medicine, Chiang Mai University, Chiang Mai, Thailand for supporting and providing access to their computing resources.

## Author Contributions

**Conceptualization:** Kanchanok Kodchakorn, Prachya Kongtawelert.

**Data curation:** Kanchanok Kodchakorn, Prachya Kongtawelert.

**Formal analysis:** Kanchanok Kodchakorn.

**Funding acquisition:** Prachya Kongtawelert.

**Investigation:** Kanchanok Kodchakorn.

**Methodology:** Kanchanok Kodchakorn.

**Project administration:** Kanchanok Kodchakorn.

**Resources:** Prachya Kongtawelert.

**Supervision:** Prachya Kongtawelert.

**Validation:** Kanchanok Kodchakorn.

**Visualization:** Kanchanok Kodchakorn.

**Writing – original draft:** Kanchanok Kodchakorn.

**Writing – review & editing:** Kanchanok Kodchakorn, Prachya Kongtawelert.

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
