## [Decision Letter · Decision Letter 0]

5 Sep 2022

PONE-D-22-17121Molecular dynamics study on strengthening behaviour of the Delta and Omicron SARS-CoV-2 Spike RBD improved receptor-binding affinityPLOS ONE

Dear Dr. Kodchakorn

Thank you for submitting your manuscript to PLOS ONE. After careful consideration, we feel that it has merit but does not fully meet PLOS ONE’s publication criteria as it currently stands. Therefore, we invite you to submit a revised version of the manuscript that addresses the points raised during the review process.

Your manuscript has been reviewed by experts in the field. Reviewers have raised valid questions and have suggestions to improve on the manuscript. I ask that you address their critiques point by point. Importantly, I ask that the statistical analysis of the work is appropriate and rigorous and that you appropriately refer to recent studies in the field.

We look forward to receiving your revised manuscript.

Kind regards,

Nicolas N. Nassar, Ph.D.

Academic Editor

PLOS ONE

Journal Requirements:

“This work was supported by grants from Thailand Excellence Center for Tissue Engineering and Stem Cells (P.K.), Department of Biochemistry, Faculty of Medicine, and Post-Doctoral Fellowship (R000030567) by CMU Presidential Scholarship (K.K) Chiang Mai University, Chiang Mai, Thailand.”

Reviewers' comments:

Reviewer's Responses to Questions

**Comments to the Author**

1. Is the manuscript technically sound, and do the data support the conclusions?

Reviewer #1: Yes

Reviewer #2: No

2. Has the statistical analysis been performed appropriately and rigorously? 

Reviewer #1: No

Reviewer #2: N/A

3. Have the authors made all data underlying the findings in their manuscript fully available?

Reviewer #1: Yes

Reviewer #2: Yes

4. Is the manuscript presented in an intelligible fashion and written in standard English?

Reviewer #1: Yes

Reviewer #2: No

5. Review Comments to the Author

Reviewer #1: Kodchakorn et al have computationally studied the SARS-CoV2 receptor binding domain interaction with ACE2. The authors examine the interaction of the wild-type virus, along with the Omicron, Delta, and India variants to understand how the substitutions have altered the viral infectivity and interactions with ACE2. In this thoughtful study, they went on to screen some of the residues to determine which interactions play an important role in binding. This information is very important, as we continue to battle the infections caused by the virus. Understanding the interactions could lead to important drug discoveries. This article could be of interest to a broad audience.

I thought that the writing could be improved to enhance readability. I appreciate the authors have included detailed methodology within the methods section. However, abbreviated information should be provided throughout the results and discussion to better understand what questions they are trying to answer and how. For instance, lines 135-142 introduced results and calculations from simulations, but the running of simulations wasn’t mentioned. Similarly, the discussion of alanine mutants in lines 148-150 introduced many questions. How were the residues selected for mutation as the entire RBD was not mutated but more residues than those shown in S1 were mutated? Were the alanines substituted in the original structure and subsequent simulations run or were they substituted into the final structure and the DDG calculated?

In figure S1 it might be helpful to include the RBD resolved in the absence of ACE2. My understanding was that the authors suggested a conformational change was not induced upon binding, but a comparison structure isn’t provided or shown. This is always a concern, since often proteins will change their configurations when binding to ligands or other interactors.

In Fig4 B-D – what do the dots represent? Are these from unique simulations? A statistical analysis should be run to determine whether the differences are statistically significant and the data presented.

The authors should also consider showing some measures of protein/complex stability (such as RMSD, RMSF) during the production runs, even if in the supplementary material. Given the number of mutations introduced for the variants and in the alanine scans, seeing that the systems are stable enhances confidence in the results presented.

Minor comments:

The minimization details should also be included in the methodology section.

The white blocks in the grey and red regions of figure 1 should be explained or removed.

It is difficult to compare the similar H-bonds across the variants in Table 2. Perhaps the table could list the occupancy for the mutants side by side or implement some color code that would shade the shared H-bonds a similar color in each variant. Also, frames as it stands is not meaningful. Perhaps it should be changed to simulation time.

Reviewer #2: Language editing is required.

"The variants of India (B.1.617.1), Delta (B.1.617.2), and Omicron (B.1.1.529) are obtained to evaluate whether or not naturally occurring mutations have strengthened viral infectivity"

variants of India (B.1.617.1) or Kappa variant????

The major changes in the binding interaction with ACE2 ... positions that almost mutated in the S-RBD region of each variant.

Rewrite this sentence

Authors are suggested to add multiple antibodies that recognize different epitopes and analyze the differences in

their binding activity to SARS-CoV-2 variants.

Poor literature search and discussion without experimental validations.

The hydrogen bonding presented in black dashed line.......occupancy (see S2 Fig). check the sentence

More Principal component analyses to be performed.

For how much time The MD simulation was performed. More analysis need to be performed for better justification.

Several important and recent studies should be corelated with your findings. The same are suggested to be cited as well

Khan et al. Impact of the Double Mutants on Spike Protein of SARS-CoV-2 B.1.617 Lineage on the Human ACE2 Receptor Binding: A Structural Insight. Viruses 2021, 13, 2295.

Tianet al. N501Y mutation of spike protein in SARS-CoV-2 strengthens its binding to receptor ACE2. Elife, 10, e69091.

Fratev, F. N501Y and K417N mutations in the spike protein of SARS-CoV-2 alter the interactions with Both hACE2 and human-derived antibody: a free energy of perturbation retrospective study. Journal of Chemical Information and Modeling, 61(12), 6079-6084.

Khan et al. Deciphering the impact of mutations on binding efficacy of SARS-CoV-2 Omicron and Delta variants with human ACE2 receptor. Frontiers in Chemistry, 566.

6. PLOS authors have the option to publish the peer review history of their article (what does this mean?). If published, this will include your full peer review and any attached files.

Reviewer #1: No

Reviewer #2: No

---

## [Author Response · Author response to Decision Letter 0]

10 Oct 2022

Response to Reviewer-1’s comments:

Reviewer #1: Kodchakorn et al have computationally studied the SARS-CoV2 receptor binding domain interaction with ACE2. The authors examine the interaction of the wild-type virus, along with the Omicron, Delta, and India variants to understand how the substitutions have altered the viral infectivity and interactions with ACE2. In this thoughtful study, they went on to screen some of the residues to determine which interactions play an important role in binding. This information is very important, as we continue to battle the infections caused by the virus. Understanding the interactions could lead to important drug discoveries. This article could be of interest to a broad audience.

I thought that the writing could be improved to enhance readability. I appreciate the authors have included detailed methodology within the methods section.

Response: Thank you very much for your suggestions. The detailed methodologies have been added in “Materials and methods” section, as marked in red letter, lines 123-258.

However, abbreviated information should be provided throughout the results and discussion to better understand what questions they are trying to answer and how. For instance, lines 135-142 introduced results and calculations from simulations, but the running of simulations wasn’t mentioned. 

Response: Reviewer is right. As the reviewers’ suggestions, the more details have been mentioned and added in each part of the “Results and discussion” section, as marked in red letter.

Similarly, the discussion of alanine mutants in lines 148-150 introduced many questions.

Response: As reviewer’s suggestions, the discussion of alanine mutants was revised for more details as marked in red letter, lines 308-320.

How were the residues selected for mutation as the entire RBD was not mutated but more residues than those shown in S1 were mutated?

Response: We performed all amino acid residues located on the spike RBD region (residues 438-505) for the alanine mutation analysis, that may directly affect the ACE2 binding. That meant why more residues were mutated than those shown in Fig S1. Furthermore, only significant relative binding energy in the mutations were presented as illustrated in Fig 2.

Were the alanines substituted in the original structure and subsequent simulations run or were they substituted into the final structure and the DDG calculated?

Response: Each mutated amino acid of S-RBD protein was directly replaced in the final structure of the wild-type S-RBD protein (pdb file). Then, the corresponding topology file in each alanine substituted structure was built and used to calculate the binding free energy by MM-PBSA method for comparison to the wild-type MD simulation as the trajectories to be analyzed. That is to say, only one mutation can be performed during a single calculation.

In figure S1, it might be helpful to include the RBD resolved in the absence of ACE2. My understanding was that the authors suggested a conformational change was not induced upon binding, but a comparison structure isn’t provided or shown. This is always a concern, since often proteins will change their configurations when binding to ligands or other interactors.

Response: Thank you very much for your suggestions. Please correct me if I am wrong. Do you mention on a section of the alanine scanning calculation according to Fig S1? We measured the mutation effects in the key hotspot subset on a site that directly contact ACE2 in the wild-type S-RBD structure based on the alanine scanning application. The mutations that enhance affinity are notable at s-RBD regions, i.e., N487, Q493, and Q498. Although these residues are involved in a dense framework of polar contacts with ACE2, our measurements show there is substantial plasticity in this interaction, as mutations that reduce the polar properties of these residues can enhance binding affinity.

 According to these key residues, in Fig 3, we described the final structure of the wild-type system to highlight the potential role that should be conserved as true hot spots. For Fig. S1, therefore, the comparison structure may not be able to include the RBD resolved in the absence of ACE2 because of their calculation method. However, the comparison structure was mentioned and discussed in detail for conformational change in “Structural analysis and stability of the wild and mutant variants SARS-CoV-2 spike protein complexes” part, as marked in red letter, lines 382-435.

In Fig4 B-D – what do the dots represent? Are these from unique simulations? A statistical analysis should be run to determine whether the differences are statistically significant and the data presented.

Response: The dots in Fig 5b illustrates an of hydrogen bond occupancies (%) with more than 10% occupancy formed by residue 493 in each S-RBD variant at ACE2 binding interface, while Fig 5d represents a number of intermolecular hydrogen bonds (in frame structures) between residue 493 of each variant and the salt-bridge residues of the ACE2.

 All data plots were obtained from the hydrogen bond analysis over production phase MD simulations.

 As reviewer’s suggestions, the statistical analysis was performed and mentioned in the revised Fig 5.

The authors should also consider showing some measures of protein/complex stability (such as RMSD, RMSF) during the production runs, even if in the supplementary material. Given the number of mutations introduced for the variants and in the alanine scans, seeing that the systems are stable enhances confidence in the results presented.

Response: As reviewer’s suggestions, the details of RMSD and RMSF plots, Fig. 4, for complex stability were added in “Structural analysis and stability of the wild and mutant variants SARS-CoV-2 spike protein complexes” part, “Results and discussion” section, as marked in red letter, lines 382-435.

Minor comments:

The minimization details should also be included in the methodology section.

Response: As reviewer’s suggestions, the information of the minimization method was added in the “Materials and methods” section, as marked in red letter, lines 158-162.

The white blocks in the grey and red regions of figure 1 should be explained or removed.

Response: As reviewer’s suggestions, the white blocks in the gray and red regions of Fig 1 were erased to avoid the missing the information.

It is difficult to compare the similar H-bonds across the variants in Table 2. Perhaps the table could list the occupancy for the mutants side by side or implement some color code that would shade the shared H-bonds a similar color in each variant. Also, frames as it stands is not meaningful. Perhaps it should be changed to simulation time.

Response: Table 2 was revised. The similar hydrogen bond formation of the ACE2-bound S-RBD complex obtained by the wild-type and the mutated systems were represented in bold style for reasonable comparison. The frame value was also erased as reviewer’s suggestion.

Response to Reviewer-2’s comments:

Reviewer #2: Language editing is required.

Response: The manuscript has been proofread as reviewer’s suggestions.

"The variants of India (B.1.617.1), Delta (B.1.617.2), and Omicron (B.1.1.529) are obtained to evaluate whether or not naturally occurring mutations have strengthened viral infectivity", variants of India (B.1.617.1) or Kappa variant?

Response: The reviewer is right. As reviewer’s suggestions, the word “India” was replaced by the “Kappa” as marked in red letter.

The major changes in the binding interaction with ACE2 ... positions that almost mutated in the S-RBD region of each variant. Rewrite this sentence

Response: This section has been erased according to the journal formatting guideline.

Authors are suggested to add multiple antibodies that recognize different epitopes and analyze the differences in their binding activity to SARS-CoV-2 variants.

Poor literature search and discussion without experimental validations.

Response: As reviewer’s suggestions, more details were added and discussed related to the experimental studies in the “Results and discussion” section, as marked in red letter.

The hydrogen bonding presented in black dashed line.......occupancy (see S2 Fig). check the sentence

Response: As reviewer’s suggestions, the sentence was revised as marked in red letter, line 514 and in the legend of Fig 5.

More Principal component analyses to be performed. For how much time, The MD simulation was performed. More analysis needs to be performed for better justification.

Response: In this work, 50 ns production phase MD simulation was carried out on all complex systems.

 As reviewer’s suggestions, more detailed analysis was added and discussed in the “Results and discussion” section, as marked in red letter.

 “Structural analysis and stability of the wild and mutant variants SARS-CoV-2 spike protein complexes” part, lines 382–435.

 “Dynamic cross-correlation matrix and conformational flexibility analysis” part, lines 690–759.

Several important and recent studies should be corelated with your findings. The same are suggested to be cited as well.

Khan et al. Impact of the Double Mutants on Spike Protein of SARS-CoV-2 B.1.617 Lineage on the Human ACE2 Receptor Binding: A Structural Insight. Viruses 2021, 13, 2295.

Tianet al. N501Y mutation of spike protein in SARS-CoV-2 strengthens its binding to receptor ACE2. Elife, 10, e69091.

Fratev, F. N501Y and K417N mutations in the spike protein of SARS-CoV-2 alter the interactions with Both hACE2 and human-derived antibody: a free energy of perturbation retrospective study. Journal of Chemical Information and Modeling, 61(12), 6079-6084.

Khan et al. Deciphering the impact of mutations on binding efficacy of SARS-CoV-2 Omicron and Delta variants with human ACE2 receptor. Frontiers in Chemistry, 566.

Response: Thank you very much for your suggestions. The recent literatures were added and the details were discussed for reasonable results as your suggestion in the “Results and discussion” section, as marked in red letter.

---

## [Decision Letter · Decision Letter 1]

3 Nov 2022

Molecular dynamics study on the strengthening behavior of Delta and Omicron SARS-CoV-2 spike RBD improved receptor-binding affinity

PONE-D-22-17121R1

Dear Dr. Kodchakorn,

We’re pleased to inform you that your manuscript has been judged scientifically suitable for publication and will be formally accepted for publication once it meets all outstanding technical requirements.

Kind regards,

Nicolas N. Nassar, Ph.D.

Academic Editor

PLOS ONE

Additional Editor Comments (optional):

Reviewers' comments:

Reviewer's Responses to Questions

**Comments to the Author**

1. If the authors have adequately addressed your comments raised in a previous round of review and you feel that this manuscript is now acceptable for publication, you may indicate that here to bypass the “Comments to the Author” section, enter your conflict of interest statement in the “Confidential to Editor” section, and submit your "Accept" recommendation.

Reviewer #1: All comments have been addressed

Reviewer #2: All comments have been addressed

2. Is the manuscript technically sound, and do the data support the conclusions?

Reviewer #1: Yes

Reviewer #2: Yes

3. Has the statistical analysis been performed appropriately and rigorously? 

Reviewer #1: Yes

Reviewer #2: Yes

4. Have the authors made all data underlying the findings in their manuscript fully available?

Reviewer #1: Yes

Reviewer #2: Yes

5. Is the manuscript presented in an intelligible fashion and written in standard English?

Reviewer #1: Yes

Reviewer #2: (No Response)

6. Review Comments to the Author

Reviewer #1: The authors have done a nice job at addressing all reviewer comments. The additional data and data analysis strengthen their findings. I have no additional concerns.

Reviewer #2: (No Response)

7. PLOS authors have the option to publish the peer review history of their article (what does this mean?). If published, this will include your full peer review and any attached files.

Reviewer #1: No

Reviewer #2: No

---

## [Editor Report · Acceptance letter]

7 Nov 2022

PONE-D-22-17121R1 

Molecular dynamics study on the strengthening behavior of Delta and Omicron SARS-CoV-2 spike RBD improved receptor-binding affinity 

Dear Dr. Kodchakorn:

I'm pleased to inform you that your manuscript has been deemed suitable for publication in PLOS ONE. Congratulations! Your manuscript is now with our production department. 

Kind regards, 

on behalf of

Dr. Nicolas N. Nassar 

Academic Editor

PLOS ONE